

# Traveling discontinuity at the quantum butterfly front

Camille Aron[1,2]⋆, Éric Brunet[1]† and Aditi Mitra[3]‡

**1** Laboratoire de Physique de l'École Normale Supérieure, ENS, Université PSL,
CNRS, Sorbonne Université, Université Paris Cité, F-75005 Paris, France
**2** Institute of Physics, École Polytechnique Fédérale de Lausanne (EPFL),
CH-1015 Lausanne, Switzerland
**3** Center for Quantum Phenomena, Department of Physics, New York University,
726 Broadway, New York, NY, 10003, USA

⋆ aron@ens.fr , † eric.brunet@ens.fr ,
‡ aditi.mitra@nyu.edu

## Abstract

We formulate a kinetic theory of quantum information scrambling in the context of a paradigmatic model of interacting electrons in the vicinity of a superconducting phase transition. We carefully derive a set of coupled partial differential equations that effectively govern the dynamics of information spreading in generic dimensions. Their solutions show that scrambling propagates at the maximal speed set by the Fermi velocity. At early times, we find exponential growth at a rate set by the inelastic scattering. At late times, we find that scrambling is governed by shock-wave dynamics with traveling waves exhibiting a discontinuity at the boundary of the light cone. Notably, we find perfectly causal dynamics where the solutions do not spill outside of the light cone.

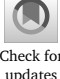

# 1 Introduction

Quantum information scrambling is the mechanism by which localized information in an extended closed quantum many-body system with local interactions flows to non-local degrees of freedom, becoming practically irretrievable. In practice, these information dynamics can be conveniently probed by means of out-of-time-ordered correlators (OTOCs) such as squared commutators of operators inserted at different space-time points, *e.g.* $\mathcal{C}(t,\boldsymbol{x}) := \langle [\mathcal{O}(t,\boldsymbol{x}), \mathcal{O}(0,0)]^2 \rangle$. The effective loss of information is characterized by (i) a ballistic spread of information, often dubbed as the "quantum butterfly effect", (ii) a growth regime, reminiscent of the exponential separation between nearby trajectories in classical chaotic systems, (iii) a purely quantum saturation regime beyond a scrambling time $t_*$.

The ballistic spread of quantum information has been firmly established on the basis of the Lieb-Robinson bound [1]. The causal light-cone structure, with a wavefront traveling at a model-dependent butterfly velocity $v_{\mathrm{B}}$, was confirmed in a wide variety of models, from non-interacting $1d$ systems to holographic models. Inside the light cone, the existence of a clearly delineated exponential growth regime is only expected for semiclassical or large-$N$ models: $\mathcal{C}(t,\boldsymbol{x}) \sim \exp[\lambda_{\mathrm{L}}(t - t_* - |\boldsymbol{x}|/v_{\mathrm{B}})]$ where $\lambda_{\mathrm{L}}$ is the Lyapunov exponent. For truly quantum systems, the rapid growth concentrated near the light cone boundary is understood to be set by model-dependent microscopic scales, and significant efforts were made to compute the particular shape of the butterfly front in a variety of models.

Wavefronts described by power laws, sometimes oscillatory, were found in free and integrable models [2, 3]. Sharp wavefronts were found in interacting spin chains [4], non-integrable systems with diffusive transport [5, 6], as well as large-$N$ or semiclassical models [7, 8] and holographic models [9, 10]. Interestingly, random unitary circuits without conserved quantities, *i.e.* in the absence of diffusive transport, were found to develop broad fronts controlled by a diffusively growing length scale [11–14]. Notably, several works [15–18] have

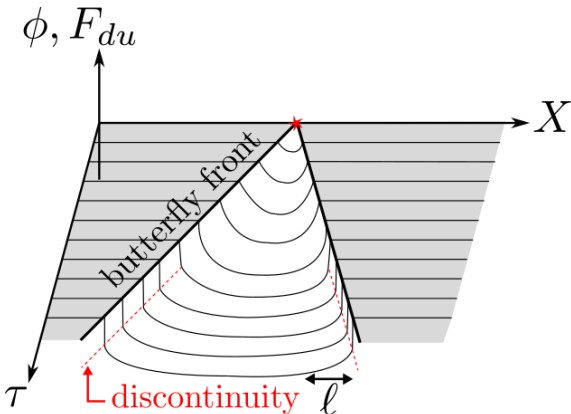

Figure 1: Butterfly effect in the clean interacting metal defined in Eqs. (5) and (22): after a small and localized perturbation at time $\tau = 0$, the space-time dynamics of the inter-world distribution function $F_{du}$ defined in Eq. (15) follows a causal light-cone structure with a front traveling at the maximal butterfly velocity $v_{\mathrm{B}} = v_{\mathrm{F}}/\sqrt{d}$ where $v_{\mathrm{F}}$ is the Fermi velocity and $d$ the spatial dimensions. Here, we sketch $\phi(\tau, X)$, the component of $F_{du}$ averaged over the Fermi surface, to be introduced in Eq. (31). At early times, the exponential growth regime is governed by the inelastic scattering rate. In the late-time limit, scrambling is governed by shock-wave dynamics with the traveling front developing a distinct discontinuity followed by an exponential decay on the scale of the mean free path $\ell$ set by the electronic interaction.

pointed towards an effective description of the front in terms of partial differential equations belonging to the class of the Fisher Kolmogorov-Petrovsky–Piskunov (FKPP) equation [19,20] which is known to exhibit traveling wave solutions (see Ref. [21] and references therein).

In this manuscript, we address the dynamics and the geometry of the wavefront for a locally interacting system in the vicinity of a continuous classical phase transition corresponding to the spontaneous breaking of the symmetry associated with a conserved quantity. Practically, we consider a paradigmatic model of interacting electrons where the interactions are due to strong superconducting fluctuations. The long wavelength fluctuations close to criticality produce a separation of scales which we leverage to derive analytic results. How the results get modified on moving away from criticality is transparent in our derivation and our approach can be systematized to address other near-critical quantum many-body systems.

We compute the OTOCs by means of an augmented Keldysh formalism, the so-called many-world formalism, which was originally proposed in Ref. [22] and recently used to derive kinetic equations for the spreading of quantum information in fermionic interacting systems including electron-phonon scattering, electrons-impurity scattering, as well as electron-electron scattering [15]. The augmented Keldysh formalism has been used in other recent works as well [23–26]. This formalism can be conveniently harnessed to the standard field-theoretic concepts, tools, and approximation schemes that have been developed over the years in condensed matter theory. Here, we treat the interaction between the electrons and the superconducting fluctuations by means of the random-phase approximation (RPA) in the particle-particle channel.

We carefully derive an effective description of the spreading of quantum information in terms of a set of coupled partial differential equations which do not belong to the FKPP class. Notably, we find wavefronts that are discontinuous at the light cone boundary and that do not feature exponentially small tails ahead of the front. This strictly causal structure is unlike what is found in evolutions of the FKPP class, and more generally of equations with diffusive terms.

**Summary and main results**

The paper is organized as follows. In Sect. 2, we quickly review the many-world formalism which is used to compute OTOCs and access quantum chaotic features of many-body systems. It generalizes the standard Keldysh formalism by studying two replicas of the theory, the so-called worlds, which are only correlated through their initial conditions. In particular, we introduce the inter-world distribution function $F_{\alpha\beta}(\omega, \boldsymbol{k}; t, \boldsymbol{x})$, where $\alpha \neq \beta$ are the world indices, which quantifies the amount of correlation between the two worlds.

In Sect. 3, we introduce a paradigmatic model describing interacting electrons close to a superconducting transition. We avoid the technical challenges of approaching the critical point from within the superconducting phase and work in the normal phase where no long-range order develops. We derive the corresponding kinetic equation for the inter-world distribution which is shown to be of the form

$$\partial_t F_{\alpha\beta} + \boldsymbol{v_k} \cdot \boldsymbol{\nabla_x} F_{\alpha\beta} = I_{\alpha\beta}[F_{\alpha\beta}, F_{\beta\alpha}],$$

with the non-linear collision integral $I_{\alpha\beta}$ given in Eq. (30).

In Sect. 4, we propose and numerically validate an ansatz for $F_{\alpha\beta}$ leading to a simplified set of non-linear partial differential equations (PDEs) involving two fields $\phi(t, \boldsymbol{x})$ and $\boldsymbol{\phi}_1(t, \boldsymbol{x})$. These are the first terms of the partial-wave expansion in momentum space of $F_{\alpha\beta}(\omega, \boldsymbol{k}; t, \boldsymbol{x})$ evaluated on-shell and at the Fermi surface, *i.e.* at $\omega = \epsilon_{\boldsymbol{k}}$ and $k \to k_{\mathrm{F}}$. In terms of dimensionless quantities $\boldsymbol{X}$ for space and $\tau$ for time, the PDEs read

$$\begin{cases} \partial_\tau \phi + \boldsymbol{\nabla_X} \cdot \boldsymbol{\phi}_1 = \phi(\phi^2 - 1), \\ \partial_\tau \boldsymbol{\phi}_1 + \boldsymbol{\nabla_X} \phi = \boldsymbol{\phi}_1(\gamma\phi^2 - 1), \end{cases}$$

where the parameter $\gamma$ effectively encodes the distance from the superconducting phase transition: $\gamma = 1$ corresponds to criticality and $0 < \gamma < 1$ corresponds to the off-critical regime in the normal phase.

In Sect. 5, starting from a generic localized initial perturbation, we analytically solve for the dynamics of $\phi$ and $\boldsymbol{\phi}_1$, in any dimension $d$, at criticality as well as away from criticality. The results are sketched in Fig. 1. The relaxation of the inter-world distribution is found to strictly occur within a light cone growing from the initial perturbation at a constant butterfly velocity $v_{\mathrm{B}} = v_{\mathrm{F}}/\sqrt{d}$ where $v_{\mathrm{F}}$ is the Fermi velocity. We work out the early-time dynamics with an exponential growth of scrambling which is controlled by the inelastic scattering rate. Importantly, in the late-time regime, we find that scrambling is governed by shock-wave dynamics, with a traveling wave that develops a discontinuous radial front of the form $F_{\alpha\beta} \sim f_+\left(\frac{|\boldsymbol{x}| - v_{\mathrm{B}} t}{\ell/\sqrt{d}}\right)$ which extends over a length scale set by the mean free path $\ell$ related to the scattering of the electrons by superconducting fluctuations. Inside the light cone, $f_+$ dies off exponentially away from its boundary ($|\boldsymbol{x}| - v_{\mathrm{B}} t < 0$), and $f_+ = 1$ outside the light cone ($|\boldsymbol{x}| - v_{\mathrm{B}} t > 0$). Notably, we find that $f_+$ is discontinuous precisely at the boundary ($|\boldsymbol{x}| - v_{\mathrm{B}} t = 0$). We also work out explicitly the exponential falloff governing the approach to the saturation regime within the bulk of the light cone.

We conclude in Sect. 6 by discussing the relations of our results to previous works and by giving future directions.

## 2 Many-world formalism

### 2.1 Motivations and general idea

Let us first motivate the use of the so-called many-world formalism and review its basic functioning. Dynamical signature of quantum chaos can be found in OTOCs of the type (we set

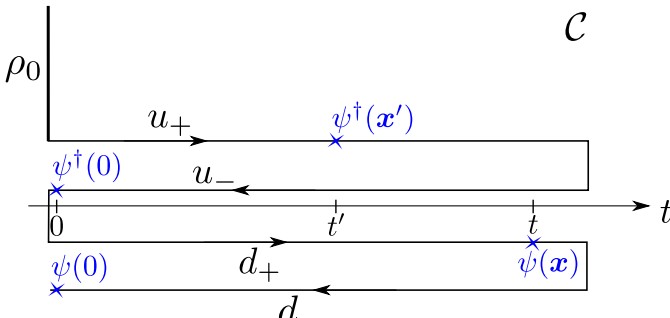

Figure 2: Two-world Keldysh contour $\mathcal{C}$: the theory is replicated into an "up" world dynamics, marked by the index $u$, and the "down" world dynamics marked by $d$. The location of the operators $\psi$ and $\psi^\dagger$ correspond to the OTOC in Eq. (1).

$\hbar = 1$)

$$\text{Tr}\left[\psi(0)\,\mathrm{e}^{\mathrm{i}Ht}\psi(\boldsymbol{x})\mathrm{e}^{-\mathrm{i}Ht}\,\psi^\dagger(0)\,\mathrm{e}^{\mathrm{i}Ht'}\psi^\dagger(\boldsymbol{x}')\,\mathrm{e}^{-\mathrm{i}Ht'}\rho_0\right],\tag{1}$$

where $\rho_0$ is the initial density matrix at time $t = 0$ which is normalized as $\text{Tr}\,\rho_0 = 1$, $H$ is the Hamiltonian generating the dynamics, and $\psi(\boldsymbol{x})$ is a local operator at position $\boldsymbol{x}$. We have in mind fermionic annihilation/creation operators, but the discussion can be easily adapted to the bosonic case. Here, the four operators $\psi(0)$, $\psi(\boldsymbol{x})$, $\psi^\dagger(0)$, and $\psi^\dagger(\boldsymbol{x}')$ are computed in a non-ordered time sequence. Like many of the diagnostics of quantum chaos, the convoluted time structure of OTOCs makes them computable objects which do not, however, directly correspond to physical observables. This is in contrast to standard retarded correlators which correspond to response functions to physical perturbations. Consequently, the computation of OTOCs requires modifying the standard non-equilibrium Green's function approach to cope with the out-of-time ordering.

Here, we quickly review the many-world formalism which generalizes the standard Schwinger-Keldysh formalism defined on the two-fold Baym-Kadanoff contour to a formalism on a four-fold contour, suitable to compute four-point OTOCs. This was originally introduced in Ref. [22] and we refer the reader to Ref. [15] for a detailed presentation which we simply follow. The OTOC in Eq. (1) involves two forward and two backward time-evolution operators. Therefore, following the standard Schwinger-Keldysh construction [27], this yields the four-fold time-ordered contour $\mathcal{C}$ depicted in Fig. 2. The forward (backward) branches are labeled by an index $a = +$ ($a = -$). The first two branches are said to belong to the "up world" and are labeled by the index $\alpha = u$. The two other branches, posterior on the contour, correspond to the so-called "down world", and are labeled by $\alpha = d$. Notably, the up and down worlds are identical replicas of the same theory, involving the same Hamiltonian.

In that language, the OTOC in Eq. (1) can be rewritten as

$$\left\langle \mathrm{T}_{\mathcal{C}}\,\psi_d^-(0,0)\psi_d^+(t,\boldsymbol{x})\psi_u^{-\dagger}(0,0)\psi_u^{+\dagger}(t',\boldsymbol{x}')\right\rangle,\tag{2}$$

where $\mathrm{T}_{\mathcal{C}}$ is the time-ordering operator on the contour $\mathcal{C}$, the operators $\psi$ and $\psi^\dagger$ are now written in the Heisenberg picture, and $\langle\ldots\rangle := \text{Tr}\,[\ldots\rho_0]$.

In our subsequent study of the quantum butterfly effect, we shall assume equilibrium conditions: the system is initially prepared in thermal equilibrium at temperature $T$, $i.e.$ with the Gibbs state $\rho_0 \sim \mathrm{e}^{-H/T}$ where we set $k_{\mathrm{B}} = 1$, and the subsequent evolution is unitarily generated by the same Hamiltonian $H$ as the one used in the initial preparation.

It is useful to introduce the following quantities:

$$\mathrm{i}G_{\alpha\beta}^{ab}(t,\boldsymbol{x};t',\boldsymbol{x}') := \left\langle \mathrm{T}_{\mathcal{C}}\psi_\alpha^a(t,\boldsymbol{x})\psi_\beta^{b\dagger}(t',\boldsymbol{x}')\hat{S}_0\right\rangle,\tag{3}$$

with the Keldysh indices $a, b = +, -$, the world indices $\alpha, \beta = u, d$, and where the mixed operator $\hat{S}_0 := \psi_u^{-\dagger}(0,0)\psi_d^-(0,0)$ results from clubbing of the two operators in Eq. (2) that are both evaluated at time $t = 0$ but at disconnected locations on the time contour $\mathcal{C}$. The OTOC in Eq. (1) is recovered by taking $\alpha = d$ and $\beta = u$, while $a = \pm$ and $b = \pm$ can be chosen arbitrarily. If one is to interpret $iG_{\alpha\beta}^{ab}(t, \boldsymbol{x}; t', \boldsymbol{x}')$ as a two-point function rather than a four-point function, $\hat{S}_0$ has to be understood as a local modification of the initial condition which, *a priori,* acts differently on each world: one particle is added to the up world at position $\boldsymbol{x} = 0$ while one particle is removed from the down world. Other choices of operator content are possible for $\hat{S}_0$ and we shall see that the late-time inter-world dynamics depend very little on this choice. Later, we shall consider local perturbations around the identity: $\hat{S}_0 = 1 + \delta\phi_0 \, \psi_u^{-\dagger}(0,0)\psi_d^-(0,0)$ with the infinitesimal parameter $|\delta\phi_0| \ll 1$.

Importantly, an inspection of Eq. (3) for $\hat{S}_0 = 1$ shows that the intra-world Green's functions (*i.e.* $\alpha = \beta$) correspond to the standard (*i.e.* single-world) Schwinger-Keldysh two-point correlators:

$$G_{uu}^{ab} = G_{dd}^{ab} = G^{ab} . \tag{4}$$

From now on, given that the intra-world quantities are identical in both $\alpha = u, d$ worlds, we simply drop the repeated world indices, *e.g.* $G_{\alpha\alpha} \to G$, except when this obscures the meaning. Furthermore, when using the indices $\alpha\beta$, we specifically mean $\alpha \neq \beta$ unless specified otherwise.

## 2.2 Interacting fermions

For concreteness, and to set the stage for the ensuing developments, we work in the context of interacting fermions on a $d$-dimensional lattice. Naturally, this can be easily adapted to other quantum systems. We consider the generic Hamiltonian

$$H = H_0 + H_{\text{int}} , \tag{5}$$

$$H_0 = \sum_{\boldsymbol{k} \in \text{BZ}} \sum_\sigma \epsilon_{\boldsymbol{k}} c_{\boldsymbol{k}\sigma}^\dagger c_{\boldsymbol{k}\sigma} , \tag{6}$$

where $H_0$ is the non-interacting part and the interaction in $H_{\text{int}}$ depends on the specific problem at hand, see Eq. (22). The fermionic operator $c_{\boldsymbol{k}\sigma}^\dagger$ creates a electrons with spin $\sigma = \uparrow$ or $\downarrow$ ($\bar{\sigma} = \downarrow, \uparrow$) and momentum $\boldsymbol{k}$ in the Brillouin zone (BZ). $\epsilon_{\boldsymbol{k}}$ is the dispersion relation. The generalization to multi-band cases is straightforward. For simplicity, we measure electronic energies relative to the chemical potential, but a finite chemical potential $\mu$ can be included via the substitution $\epsilon_{\boldsymbol{k}} \mapsto \epsilon_{\boldsymbol{k}} - \mu$. For simplicity, we shall assume that the Fermi surface is spherical, *i.e.* $\epsilon_{\boldsymbol{k}} = 0$ when $k \to k_{\text{F}}$. Whenever this does not harm the understanding, we shall simply drop the spin indices.

## 2.3 Green's functions in the Keldysh basis

The 16 Green's functions $G_{\alpha\beta}^{ab}$ are not independent of each other and one may considerably reduce the redundancies of the formalism. On the one hand, the causal structure of the contour $\mathcal{C}$ is such that the inter-world Green's functions (*i.e.* $\alpha \neq \beta$) do not depend on the $\pm$ basis:

$$G_{\alpha\beta}^{++} = G_{\alpha\beta}^{--} = G_{\alpha\beta}^{+-} = G_{\alpha\beta}^{-+} , \quad \text{for } \alpha \neq \beta . \tag{7}$$

On the other hand, the intra-world Green's functions (*i.e.* $\alpha = \beta$) are related via [27]

$$G_{\alpha\beta}^{+-} + G_{\alpha\beta}^{-+} = G_{\alpha\beta}^{++} + G_{\alpha\beta}^{--} , \quad \text{for } \alpha = \beta . \tag{8}$$

It is therefore customary to perform a rotation from the $a = \pm$ basis to the so-called Keldysh basis and work with retarded, advanced, and Keldysh Green's functions:

$$G^R_{\alpha\beta} = G^{++}_{\alpha\beta} - G^{+-}_{\alpha\beta}, \qquad G^A_{\alpha\beta} = G^{++}_{\alpha\beta} - G^{-+}_{\alpha\beta},$$
$$G^K_{\alpha\beta} = G^{++}_{\alpha\beta} + G^{--}_{\alpha\beta}, \tag{9}$$

where $G^A$ is simply the Hermitian conjugate of $G^R$.

The intra-world Green's functions are the standard Schwinger-Keldysh correlators, solutions of the Schwinger-Dyson equations which, in thermal equilibrium and in Fourier space, read

$$G^R(\omega, \boldsymbol{k}) = \left[\omega - \epsilon_{\boldsymbol{k}} - \Sigma^R(\omega, \boldsymbol{k})\right]^{-1},$$
$$G^K(\omega, \boldsymbol{k}) = 2\mathrm{i}\, F(\omega)\, \mathrm{Im}\, G^R(\omega, \boldsymbol{k}). \tag{10}$$

$\Sigma^R$ is the retarded component of the self-energy. It is due to the interaction in $H_{\mathrm{int}}$ and can be computed diagrammatically within the standard Schwinger-Keldysh formalism. The last equality is the expression of the fermionic fluctuation-dissipation theorem with $F(\omega) := \tanh(\omega/2T)$.

Concerning inter-world Green's functions, the relation in Eq. (7) together with Eq. (9) immediately implies

$$G^R_{ud} = G^R_{du} = 0. \tag{11}$$

This expresses the fact that while intra-world physics contributes to inter-world quantities, the opposite, namely that inter-world physics contributes to intra-world quantities, is strictly forbidden.

To summarize, given that we restrict ourselves to equilibrium physics, we are to deal with only three independent Greens' functions: the standard (intra-world) retarded Green's function $G^R$ which is uniquely specified by the Hamiltonian $H$ and the temperature $T$, and the two inter-world Keldysh Green's functions $G^K_{ud}$ and $G^K_{du}$ which also depend on the choice of the operator $\hat{S}_0$. For the choice $\hat{S}_0 = 1$, the two worlds have the same thermal initial conditions, and one may check that $G^K_{\alpha\beta}$ are space- and time-translational invariant and

$$G^K_{ud}(\omega, \boldsymbol{k}) = 2\mathrm{i}[-1 + F(\omega)]\,\mathrm{Im}\, G^R(\omega, \boldsymbol{k}),$$
$$G^K_{du}(\omega, \boldsymbol{k}) = 2\mathrm{i}[+1 + F(\omega)]\,\mathrm{Im}\, G^R(\omega, \boldsymbol{k}). \tag{12}$$

However, for a generic choice of $\hat{S}_0$, $G^K_{\alpha\beta}$ is not guaranteed to be space- and time-translational invariant and it can be determined via the Schwinger-Dyson equation reading

$$G^K_{\alpha\beta}(t, \boldsymbol{x}; t', \boldsymbol{x}') = \int \mathrm{d}\boldsymbol{x}_1 \int \mathrm{d}\boldsymbol{x}_2 \int \mathrm{d}t_1 \int \mathrm{d}t_2$$
$$G^R(t - t_1, \boldsymbol{x} - \boldsymbol{x}_1)\Sigma^K_{\alpha\beta}(t_1, \boldsymbol{x}_1; t_2, \boldsymbol{x}_2,)G^A(t_2 - t', \boldsymbol{x}_2 - \boldsymbol{x}'), \tag{13}$$

where $\Sigma^K_{\alpha\beta}$ is the Keldysh component of the inter-world self-energy which can be computed diagrammatically in the many-world formalism.

## 2.4 Inter-world kinetic equation

It is useful to work in the Wigner representation

$$G^K_{\alpha\beta}(\omega, \boldsymbol{k}; t, \boldsymbol{x}) := \int \mathrm{d}\boldsymbol{x}' \int \mathrm{d}t'\, \mathrm{e}^{\mathrm{i}(\omega t' - \boldsymbol{k}\cdot\boldsymbol{x}')} G^K_{\alpha\beta}\left(t + \frac{t'}{2}, \boldsymbol{x} + \frac{\boldsymbol{x}'}{2}; t - \frac{t'}{2}, \boldsymbol{x} - \frac{\boldsymbol{x}'}{2}\right), \tag{14}$$

and parameterize $G^K_{\alpha\beta}$ in terms of the real function $F_{\alpha\beta}$:

$$G^K_{\alpha\beta}(\omega,\boldsymbol{k};t,\boldsymbol{x}) = G^R(\omega,\boldsymbol{k}) \star F_{\alpha\beta}(\omega,\boldsymbol{k};t,\boldsymbol{x}) - F_{\alpha\beta}(\omega,\boldsymbol{k};t,\boldsymbol{x}) \star G^A(\omega,\boldsymbol{k}), \qquad (15)$$

where we introduced the Moyal product $\star := \exp\left[\frac{i}{2}(\overleftarrow{\partial_\omega}\overrightarrow{\partial_t} - \overleftarrow{\boldsymbol{\nabla}_k}\cdot\overrightarrow{\boldsymbol{\nabla}_x} - \overleftarrow{\partial_t}\overrightarrow{\partial_\omega} + \overleftarrow{\boldsymbol{\nabla}_x}\overrightarrow{\boldsymbol{\nabla}_k})\right]$ where the left (right) arrow designates a derivative operator acting on the left (right) of the star symbol. In analogy to the standard (intra-world) electronic distribution function $F$, $F_{\alpha\beta}$ is dubbed the inter-world distribution function. One has $F_{ud}(\omega,\boldsymbol{k};t,\boldsymbol{x}) \in [-2,0]$ and $F_{du} \in [0,2]$. Assuming that the variations of $F_{\alpha\beta}$ occur on scales much larger than the microscopic scales involved in $G^R$, we may work in the so-called quasi-classical or gradient approximation which consists in truncating the derivative expansion to its leading terms.

The interworld distribution $F_{\alpha\beta}$ gives access to the information scrambling as it is related to the original OTOC in Eq. (2). Let us briefly outline this connection. At late times, we expect the decoupling

$$\begin{aligned}
\left\langle T_{\mathcal{C}}\, \psi^-_d(0,0)\psi^+_d(t,\boldsymbol{x})\psi^{-\dagger}_u(0,0)\psi^{+\dagger}_u(t',\boldsymbol{x}')\right\rangle \\
\simeq (1-n_0)\,iG^{++}_{du}(t,\boldsymbol{x};t',\boldsymbol{x}') \propto G^K_{du}(t,\boldsymbol{x};t',\boldsymbol{x}') \propto F_{du}(t,\boldsymbol{x};t',\boldsymbol{x}'), \qquad (16)
\end{aligned}$$

with $n_0 := \text{Tr}\left[\psi^\dagger(0)\psi(0)\rho_0\right]$ and where the interworld quantities are computed from the interworld Schwinger-Dyson equations in the presence of a local perturbation to the correlated-world solution at $\boldsymbol{x} = 0$ and time $t = 0$.

Massaging Eq. (15) by acting on both sides with the inverse of $G^R$ and $G^A$, and using the Dyson equations (10) and (13), one derives the kinetic equation on $F_{\alpha\beta}$ which is analogous to the standard (intra-world) kinetic equation:

$$\left[\partial_t + \boldsymbol{v}_k \cdot \boldsymbol{\nabla}_x\right]F_{\alpha\beta}(\omega,\boldsymbol{k}) = I_{\alpha\beta}(\omega,\boldsymbol{k}), \qquad (17)$$

where the LHS corresponds to the non-interacting physics set by $H_0$, with the velocity $\boldsymbol{v}_k := \boldsymbol{\nabla}_k \epsilon_k$, and the right-hand side (RHS) is the so-called collision integral which stems from $H_{\text{int}}$ and reads

$$I_{\alpha\beta}(\omega,\boldsymbol{k}) = 2\,\text{Im}\Sigma^R(\omega,\boldsymbol{k})F_{\alpha\beta}(\omega,\boldsymbol{k}) + i\Sigma^K_{\alpha\beta}(\omega,\boldsymbol{k}). \qquad (18)$$

Let us recall that Keldysh components and collision integrals depend on space and time, namely $\boldsymbol{x}$ and $t$, through the inter-world distribution functions. However, here and from now on, we simplify the notation by dropping the explicit dependence on these objects.

As discussed above, the intra-world quantity $\Sigma^R$ cannot depend on $F_{\alpha\beta}$, however $\Sigma^K_{\alpha\beta}$ is expected to be a non-linear functional of $F_{\alpha\beta}$. The inter-world kinetic equation in Eq. (17) is therefore a non-linear partial integrodifferential equation. It has a trivial steady-state solution

$$F^{\text{uncorr}}_{ud}(\omega,\boldsymbol{k}) = F^{\text{uncorr}}_{du}(\omega,\boldsymbol{k}) = 0, \qquad (19)$$

which reflects a total loss of coherence between the two replicated worlds, and which is dubbed the "uncorrelated-world" solution. Additionally, one can easily check that the case of $\hat{S}_0 = 1$ in Eq. (12), where both worlds evolve coherently, corresponds to another steady state characterized by

$$\begin{aligned}
F^{\text{corr}}_{ud}(\omega,\boldsymbol{k}) &= -1 + \tanh(\omega/2T)\,, \\
F^{\text{corr}}_{du}(\omega,\boldsymbol{k}) &= +1 + \tanh(\omega/2T)\,,
\end{aligned} \qquad (20)$$

and which is dubbed the "correlated-world" solution. As we shall see explicitly later, the correlated-world solution at $\hat{S}_0 = 1$ is expected to be unstable against small perturbations of $\hat{S}_0$ and the only stable steady-state is the uncorrelated-world solution.

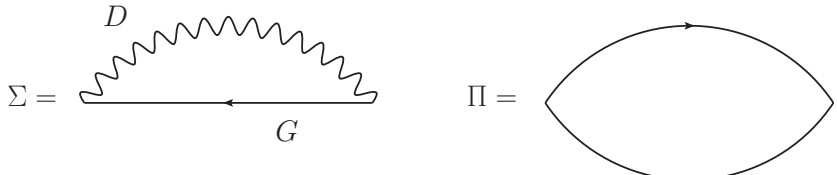

Figure 3: Superconducting self-energy $\Sigma$ and polarization bubble $\Pi$ of the model in Eq. (22) treated in the RPA scheme in the particle-particle channel. The expressions in the Keldysh formalism are given in Eqs. (25) and (24), respectively.

The aim of this manuscript is to derive and analyze the dynamics of the inter-world distribution function of a near-critical system of interacting electrons when the initial condition is of the form

$$F_{\alpha\beta}(\omega, \boldsymbol{k}; t = 0, \boldsymbol{x}) = [1 - \delta\phi_0(\boldsymbol{x})]F_{\alpha\beta}^{\text{corr}}(\omega, \boldsymbol{k}), \tag{21}$$

where $0 \leq \delta\phi_0(\boldsymbol{x}) \ll 1$ is an initial perturbation to the correlated-world solution localized on a compact support around $\boldsymbol{x} = 0$. For simplicity, we consider a perturbation that is the same for both $F_{ud}$ and $F_{du}$.

# 3  Superconducting fluctuations

## 3.1  Model

Concretely, we consider the standard Hubbard-like electron-electron interaction restricted to the particle-particle (Cooper) channel. The interacting piece of the Hamiltonian in Eq. (5) reads

$$H_{\text{int}} = U \sum_{\boldsymbol{k}\boldsymbol{k}'\boldsymbol{k}''} \sum_{\sigma} c_{\boldsymbol{k}\sigma}^{\dagger} c_{-\boldsymbol{k}+\boldsymbol{k}'\bar{\sigma}}^{\dagger} c_{\boldsymbol{k}''+\boldsymbol{k}'\bar{\sigma}} c_{-\boldsymbol{k}''\sigma}, \tag{22}$$

where $U < 0$ is an attractive interaction facilitating superconductivity. In dimensions $d \geq 2$, this model exhibits a finite-temperature phase transition towards a superconducting phase associated with the spontaneous breaking of the $U(1)$ symmetry. Here, we consider the near-critical regime, above the critical temperature where the $U(1)$ symmetry is not broken but the superconducting fluctuations are sizable.

The (intra-world) physics of this model is well understood, and we rely on standard and well-tested methods which we extend to the many-world formalism. In practice, we decouple the Hubbard interaction in the Cooper channel and obtain a theory of fermions coupled to bosonic fluctuations. The Cooperon Green's functions within RPA in the particle-particle channel, are given by [28–31]

$$
\begin{aligned}
D^R(\omega, \boldsymbol{k}) &= \left[ U^{-1} - \Pi^R(\omega, \boldsymbol{k}) \right]^{-1}, \\
D^K(\omega, \boldsymbol{k}) &= 2\mathrm{i}\, P(\omega)\, \mathrm{Im}\, D^R(\omega, \boldsymbol{k}),
\end{aligned}
\tag{23}
$$

where $\Pi^R$ is the retarded Cooper bubble and the last equality is the bosonic fluctuation-dissipation theorem with $P(\omega) := \coth(\omega/2T)$. The retarded Cooper bubble and the electronic self-energies within RPA, *i.e.* limiting ourselves to the one-loop diagrams depicted in

Fig. 3 are:

$$\mathrm{Im}\,\Pi^R(\omega, \boldsymbol{k}) = \sum_{\boldsymbol{k}'} \int \frac{\mathrm{d}\omega'}{2\pi} \mathrm{i} G^K(\omega', \boldsymbol{k}') \,\mathrm{Im}\,G^R(\omega - \omega', \boldsymbol{k} - \boldsymbol{k}'),$$

$$\Pi^K(\omega, \boldsymbol{k}) = 2\mathrm{i}\, P(\omega)\,\mathrm{Im}\,\Pi^R(\omega, \boldsymbol{k}),$$

(24)

and

$$\mathrm{Im}\,\Sigma^R(\omega, \boldsymbol{k}) = -\frac{1}{2}\sum_{\boldsymbol{k}'}\int \frac{\mathrm{d}\omega'}{2\pi}\left\{\mathrm{Im}\,D^R(\omega', \boldsymbol{k}')\mathrm{i} G^K(\omega' - \omega, \boldsymbol{k}' - \boldsymbol{k})\right.$$

$$\left. -\mathrm{i} D^K(\omega', \boldsymbol{k}')\,\mathrm{Im}\,G^R(\omega' - \omega, \boldsymbol{k}' - \boldsymbol{k})\right\},$$

(25)

$$\Sigma^K(\omega, \boldsymbol{k}) = 2\mathrm{i}\, F(\omega)\,\mathrm{Im}\,\Sigma^R(\omega, \boldsymbol{k}).$$

The real parts of the retarded components can be recovered via the Kramers–Kronig relation. A self-consistent treatment of Green's functions, self-energies, and bubbles ensures that the RPA scheme is a conserving approximation. It is known to be exact in the large $N$-limit, where $N$ is the number of electronic orbitals [28–31].

Close to criticality, the Cooperon propagator reads [28–32]

$$D^R(\omega, \boldsymbol{k}) \approx \frac{-1/\rho_{\mathrm{F}}}{r - \mathrm{i} a\omega/T + \xi^2 k^2 + \dots},$$

(26)

where $\rho_{\mathrm{F}}$ is the density of states at the Fermi energy, and the parameter $r \propto (T - T_{\mathrm{c}})/T_{\mathrm{c}}$ is the detuning from the critical point. In addition, the remaining parameters are all positive with $a \sim \mathcal{O}(1)$, $\xi^2 \sim v_{\mathrm{F}}^2/T^2$, where $v_{\mathrm{F}}$ is the Fermi velocity. At criticality $r \to 0$, the Cooperon becomes soft with diverging length scale $l \sim 1/r^\nu$ and timescale $\sim l^z$ (here $\nu = 1/2$, $z = 2$), and the propagator is singular at $\omega = k = 0$.

## 3.2 Inter-world collision integral

Let us now discuss the expressions of inter-world quantities which are necessary to compute the inter-world collision integral in Eq. (18). As we already noted, the inter-world retarded components of the Green's functions (fermionic and Cooperon), the bubbles and the self-energies simply vanish as a consequence of the fact that intra-world physics can be expressed independently of inter-world quantities: $G_{\alpha\beta}^R = D_{\alpha\beta}^R = \Pi_{\alpha\beta}^R = \Sigma_{\alpha\beta}^R = 0$ for $\alpha \neq \beta$. Within the RPA scheme, the inter-world Keldysh components read

$$D_{\alpha\beta}^K(\omega, \boldsymbol{k}) = \left|D^R(\omega, \boldsymbol{k})\right|^2 \Pi_{\alpha\beta}^K(\omega, \boldsymbol{k}),$$

(27)

$$\Pi_{\alpha\beta}^K(\omega, \boldsymbol{k}) = \frac{\mathrm{i}}{2}\sum_{\boldsymbol{k}'}\int \frac{\mathrm{d}\omega'}{2\pi}G_{\alpha\beta}^K(\omega', \boldsymbol{k}')G_{\alpha\beta}^K(\omega - \omega', \boldsymbol{k} - \boldsymbol{k}'),$$

(28)

$$\Sigma_{\alpha\beta}^K(\omega, \boldsymbol{k}) = -\frac{\mathrm{i}}{2}\sum_{\boldsymbol{k}'}\int \frac{\mathrm{d}\omega'}{2\pi}D_{\alpha\beta}^K(\omega', \boldsymbol{k}')G_{\beta\alpha}^K(\omega' - \omega, \boldsymbol{k}' - \boldsymbol{k}).$$

(29)

Altogether, this yields the inter-world kinetic equation (17) with the collision integral

$$I_{\alpha\beta}(\omega, \boldsymbol{k}) = 2\sum_{\boldsymbol{k}'\boldsymbol{k}''}\int \frac{\mathrm{d}\omega'}{2\pi}\frac{\mathrm{d}\omega''}{2\pi}|D^R(\omega', \boldsymbol{k}')|^2$$

$$\times\,\mathrm{Im}\,G^R(\omega'', \boldsymbol{k}'')\,\mathrm{Im}\,G^R(\omega' - \omega'', \boldsymbol{k}' - \boldsymbol{k}'')\,\mathrm{Im}\,G^R(\omega' - \omega, \boldsymbol{k}' - \boldsymbol{k})$$

$$\times\left\{\left[\tanh\left(\frac{\omega''}{2T}\right) + \tanh\left(\frac{\omega' - \omega''}{2T}\right)\right]\left[\coth\left(\frac{\omega'}{2T}\right) + \tanh\left(\frac{\omega - \omega'}{2T}\right)\right]F_{\alpha\beta}(\omega, \boldsymbol{k})\right.$$

$$\left. + F_{\alpha\beta}(\omega'', \boldsymbol{k}'')F_{\alpha\beta}(\omega' - \omega'', \boldsymbol{k}' - \boldsymbol{k}'')F_{\beta\alpha}(\omega' - \omega, \boldsymbol{k}' - \boldsymbol{k})\right\}.$$

(30)

We recall that we omitted the local space and time dependence of $I_{\alpha\beta}$ and $F_{\alpha\beta}$ to shorten the notations. As a sanity check, one may verify that the collision integral, and more precisely the term inside the curly brackets, vanishes at both the uncorrelated-world solution in Eq. (19) and the correlated-world solution in Eq. (20).

It is worthwhile noting that, contrary to standard intra-world collision integrals, there are no underlying conservation laws associated with the inter-world electronic distribution $F_{\alpha\beta}$ (such as number, energy, momentum conservation) that guarantee sum rules such as $\int \mathrm{d}\omega \sum_{k} I(\omega, \boldsymbol{k}) = 0$. In turn, this lack of underlying conserved quantities has important consequences on the relaxation dynamics of the inter-world distribution function. Indeed, the presence of conservation laws implies a separation of timescales and is typically synonymous with diffusive dynamics for perturbations that vary slowly enough.

## 4 Inter-world kinetics in the near-critical regime

In this Section, we propose and numerically validate an ansatz to the inter-world distribution $F_{\alpha\beta}$ which allows us to derive a much simpler version of the kinetic equation (17) with its collision integral in Eq. (30). This is done in the vicinity of the superconducting transition where a clear separation of energy scales can be made. The resulting effective description of the information scrambling dynamics consists of a set of coupled PDEs that we solve analytically in Sect. 5.

### 4.1 Partial-wave ansatz

We start by making the quasi-particle approximation: In all practical instances, $F_{\alpha\beta}(\omega, \boldsymbol{k})$ appears multiplied by the density of states $\mathrm{Im}\, G^{R}(\omega, \boldsymbol{k})$, see *e.g.* Eq. (30). When quasi-particles are well defined, with a dispersion relation $\epsilon_{\boldsymbol{k}}$, the density of states is sharply peaked around $\omega = \epsilon_{\boldsymbol{k}}$ and one may seamlessly exchange $F_{\alpha\beta}(\omega, \boldsymbol{k})$ with the on-shell quasi-particle distribution function $\tilde{F}_{\alpha\beta}(\boldsymbol{k}) := F_{\alpha\beta}(\omega = \epsilon_{\boldsymbol{k}}, \boldsymbol{k})$. From now on, we use the tilde notation to denote the on-shell prescription $\omega = \epsilon_{\boldsymbol{k}}$.

Furthermore, given that relaxation is dominated by the electronic states around the Fermi level, at energy scales (*e.g.* temperature) that are much smaller than the Fermi energy, one may focus on the distribution function close to the Fermi surface by subsequently setting $k \rightarrow k_{\mathrm{F}}$.

We now propose to simplify drastically the partial integrodifferential kinetic equation in Eq. (17) with the following ansatz:

$$\tilde{F}_{\alpha\beta}^{\mathrm{ansatz}}(t, \boldsymbol{x}; \boldsymbol{k}) = \left[ \phi(t, \boldsymbol{x}) + \boldsymbol{u}_{\boldsymbol{k}} \cdot \boldsymbol{\phi}_{1}(t, \boldsymbol{x}) \right] \tilde{F}_{\alpha\beta}^{\mathrm{corr}}(\boldsymbol{k}), \tag{31}$$

where the unit vector $\boldsymbol{u}_{\boldsymbol{k}} := \boldsymbol{k}/k$. The two fields $\phi$ and $\boldsymbol{\phi}_{1}$ can be understood as the first terms of a partial-wave expansion [27] of $\tilde{F}_{\alpha\beta}$, accounting for its isotropic and anisotropic contributions in momentum space:

$$\begin{cases} \phi(t, \boldsymbol{x}) = \dfrac{1}{S_{d-1}} \displaystyle\int \mathrm{d}\Omega_{\boldsymbol{k}} \tilde{F}_{\alpha\beta}(t, \boldsymbol{x}; \boldsymbol{k}) / \tilde{F}_{\alpha\beta}^{\mathrm{corr}}(\boldsymbol{k}) \Big|_{k \rightarrow k_{\mathrm{F}}}, \\[2ex] \boldsymbol{\phi}_{1}(t, \boldsymbol{x}) = \dfrac{d}{S_{d-1}} \displaystyle\int \mathrm{d}\Omega_{\boldsymbol{k}} \boldsymbol{u}_{\boldsymbol{k}} \tilde{F}_{\alpha\beta}(t, \boldsymbol{x}; \boldsymbol{k}) / \tilde{F}_{\alpha\beta}^{\mathrm{corr}}(\boldsymbol{k}) \Big|_{k \rightarrow k_{\mathrm{F}}}, \end{cases} \tag{32}$$

where $S_{d-1} := \int \mathrm{d}\Omega_{\boldsymbol{k}}$ is the surface area of the $d-1$-sphere with unit radius and $\mathrm{d}\Omega_{\boldsymbol{k}}$ is the elementary solid angle in the direction of $\boldsymbol{k}$. We did not include higher-order terms in the ansatz, *e.g.* of the form $u_{\boldsymbol{k}}^{i} u_{\boldsymbol{k}}^{j} \phi_{2}^{ij}$.

Let us give the rationale behind the ansatz proposed in Eq. (31). Firstly, let us note that standard (intra-world) approaches consist in perturbing the distribution function around its

equilibrium value, $F = F^{\mathrm{eq}} + \delta F$, and linearizing the collision integral accordingly. This approach relies on the fact that the equilibrium distribution $F^{\mathrm{eq}}(\omega, \boldsymbol{k}) = \tanh(\omega/2T)$ is a stable steady state of the (intra-world) kinetic equation, guaranteed by the H-theorem. The inter-world case is much different as the initial condition set by $F^{\mathrm{corr}}_{\alpha\beta}$ is unstable and one cannot propose a perturbative ansatz. This explains why $\tilde{F}^{\mathrm{corr}}_{\alpha\beta}$ appears multiplicatively in Eq. (31) and why the collision integral cannot be, *a priori*, linearized. In that regard, it is similar to the ansatz used in Ref. [15].

Secondly, close to the Fermi surface which is assumed to be spherical, the solutions $\tilde{F}^{\mathrm{corr}}_{\alpha\beta}(\boldsymbol{k})$ and $\tilde{F}^{\mathrm{uncorr}}_{\alpha\beta}(\boldsymbol{k})$ do not depend on the direction of the momentum $\boldsymbol{k}$, but only on its norm $k \approx k_{\mathrm{F}}$. This means that we aim at describing the dynamics of $\tilde{F}_{\alpha\beta}$ from a momentum-space isotropic and real-space homogeneous (unstable) solution

$$\tilde{F}^{\mathrm{corr}}(\boldsymbol{k}; t = 0, \boldsymbol{x}) \longleftrightarrow \begin{cases} \phi(t = 0, \boldsymbol{x}) = 1\,, \\ \boldsymbol{\phi}_1(t = 0, \boldsymbol{x}) = 0\,, \end{cases} \tag{33}$$

to another momentum-space isotropic and real-space homogeneous (stable) solution

$$\tilde{F}^{\mathrm{uncorr}}(\boldsymbol{k}; t \to \infty, \boldsymbol{x}) \longleftrightarrow \begin{cases} \phi(t \to \infty, \boldsymbol{x}) = 0\,, \\ \boldsymbol{\phi}_1(t \to \infty, \boldsymbol{x}) = 0\,. \end{cases} \tag{34}$$

However, as will become clear below, these dynamics can only proceed by allowing anisotropy in momentum space to develop in the transient regime towards the stable steady state. This explains why we included the anisotropic term in $\boldsymbol{\phi}_1$ which can be seen as the minimal ingredient to allow for spatial relaxation.

Thirdly, let us note that $\tilde{F}^{\mathrm{ansatz}}_{\alpha\beta}(t, \boldsymbol{x}; k)$ depends on $k$ only through $\tilde{F}^{\mathrm{corr}}_{\alpha\beta}(\boldsymbol{k})$. If this is trivially true at the correlated- and uncorrelated-world solutions, we shall see later that this is also compatible with the dynamics which does not generate extra dependence on $k$.

Finally, let us note that the fields $\phi$ and $\boldsymbol{\phi}_1$ are common to both $\tilde{F}_{ud}$ and $\tilde{F}_{du}$. This stems from our choice of initial perturbation in Eq. (21).

## 4.2 Simplified kinetic equation: coupled PDEs

We proceed by injecting the ansatz (31) in the inter-world collision integral in Eq. (30) and consistently truncating its partial-wave expansion to the two lowest orders. Because this brings further simplifications, we work in the near-critical regime of the symmetric (normal) phase where the Cooperon becomes soft. In Eq. (30), this means that the term $|D^R(\omega', \boldsymbol{k}')|^2$ diverges as $\omega' \approx 0$ and $k' \approx 0$. The details of the computation are given in Appendix A. We obtain

$$\tilde{I}_{\alpha\beta}(\boldsymbol{k}) = 2(1 - \phi^2)\big[\phi + (\boldsymbol{\phi}_1 \cdot \boldsymbol{u}_k)\big]\tilde{F}^{\mathrm{corr}}_{\alpha\beta}(\boldsymbol{k})\,\mathrm{Im}\tilde{\Sigma}^R(\boldsymbol{k})\,. \tag{35}$$

Injecting the ansatz in the kinetic equation (17), dividing by $\tilde{F}^{\mathrm{corr}}_{\alpha\beta}(\boldsymbol{k})$, we get

$$\begin{aligned} \partial_t \phi + v_k(\boldsymbol{u}_k \cdot \boldsymbol{\nabla}_x)\phi + \boldsymbol{u}_k \cdot \partial_t \boldsymbol{\phi}_1 + v_k(\boldsymbol{u}_k \cdot \boldsymbol{\nabla}_x)(\boldsymbol{u}_k \cdot \boldsymbol{\phi}_1) \\ = 2(1 - \phi^2)\big[\phi + (\boldsymbol{\phi}_1 \cdot \boldsymbol{u}_k)\big]\mathrm{Im}\tilde{\Sigma}^R(\boldsymbol{k})\,. \end{aligned} \tag{36}$$

We now project on the momentum-space isotropic and first partial-wave contributions by acting with $\frac{1}{S_{d-1}}\int \mathrm{d}\Omega_k$ and $\frac{d}{S_{d-1}}\int \mathrm{d}\Omega_k \boldsymbol{u}_k$ on both sides of the above equation. We use $\int \mathrm{d}\Omega_k u_k^i u_k^j = \delta_{ij} S_{d-1}/d$. At the Fermi surface, *i.e.* eventually setting $k \to k_{\mathrm{F}}$, we obtain the following set of coupled partial differential equations (PDEs)

$$\begin{cases} \partial_t \phi + \frac{v_{\mathrm{F}}}{d}\boldsymbol{\nabla}_x \cdot \boldsymbol{\phi}_1 = \phi(\phi^2 - 1)/\tau_{\mathrm{F}}\,, \\ \partial_t \boldsymbol{\phi}_1 + v_{\mathrm{F}}\boldsymbol{\nabla}_x \phi = \boldsymbol{\phi}_1(\gamma\phi^2 - 1)/\tau_{\mathrm{F}}\,, \end{cases} \tag{37}$$

where $v_F$ is the Fermi velocity and we defined the timescale which sets the fermionic lifetime as (recalling that the self-energy only depends on the norm of $k$)

$$\frac{1}{\tau_F} := -2\,\text{Im}\,\tilde{\Sigma}^R(k_F). \tag{38}$$

Note that the temperature dependence of the original problem enters through $\tau_F$ and $\gamma$.

$\gamma$ is a dimensionless parameter that generalizes the computation performed at criticality, for which $\gamma = 1$, to near critical regimes for which $\gamma < 1$ (see Appendix A.2). After an appropriate rescaling of space and time,

$$\tau := t/\tau_F \ \text{ and } \ X := x\sqrt{d}/(v_F \tau_F), \tag{39}$$

together with $\phi_1 \mapsto \sqrt{d}\,\phi_1$, the coupled PDEs now only involve dimensionless quantities

$$\begin{cases} \partial_\tau \phi + \nabla_X \cdot \phi_1 = \phi(\phi^2 - 1), \\ \partial_\tau \phi_1 + \nabla_X \phi = \phi_1(\gamma \phi^2 - 1). \end{cases} \tag{40}$$

Importantly, the generic $d$-dimensional case can be reduced to an effective one-dimensional case. Indeed, assuming spherically-symmetric initial conditions, we may work with the radial coordinate $r$: $\phi(\tau, r)$ and $\phi_1 = \phi_1(\tau, r)u_r$ at all times. The coupled PDEs now read

$$\begin{cases} \partial_\tau \phi + \partial_r \phi_1 = -\frac{d-1}{r}\phi_1 + \phi(\phi^2 - 1), \\ \partial_\tau \phi_1 + \partial_r \phi = \phi_1(\gamma \phi^2 - 1). \end{cases} \tag{41}$$

The set of PDEs in (37) and the following expressions in Eqs. (40), (41) are one of the main results of this manuscript. Overall, this represents a considerable simplification from the original partial integrodifferential kinetic equation (17) governing the dynamics of the inter-world distribution function $F_{\alpha\beta}(\omega, k; t, x)$ with the collision integral in Eq. (30).

To provide a first intuitive understanding of the previous set of PDEs, let us briefly neglect spatial inhomogeneities of $\phi$ and $\phi_1$, and work in $d = 1$. The equation on $\phi(t)$ becomes an autonomous first-order ODE, reading

$$\partial_\tau \phi = \phi(\phi^2 - 1) = -V'(\phi). \tag{42}$$

This is a gradient descent in the potential $V(\phi) = \frac{1}{2}\phi^2 - \frac{1}{4}\phi^4$. Reinstating the original units, the rate of escape from the correlated-world solution at the unstable extremum $\phi = 1$, to the uncorrelated-world solution at the global minimum $\phi = 0$ is $t_* := -\frac{1}{2}\tau_F \log \delta\phi_0$ where $\delta\phi_0 := 1 - \phi(0) \ll 1$. At early times, the growth of the perturbation is exponential,

$$1 - \phi(t \ll t_*) \simeq \exp[2(t - t_*)/\tau_F], \tag{43}$$

while it saturates at late times,

$$\phi(t \gg t_*) \simeq \frac{1}{\sqrt{2}}\exp[-(t - t_*)/\tau_F] \to 0. \tag{44}$$

Again, the decoupling of $\phi_1$ in such a spatially homogeneous setting is evidence that the momentum anisotropy captured by $\phi_1$ is a minimal ingredient necessary to allow for spatial propagation of the relaxation from the correlated-world solution to the uncorrelated-world solution. This will be the topic of Sect. 5.

In the general case, *i.e.* in the presence of spatial inhomogeneities, it is instructive to compare the inter-world situation to the (standard) intra-world kinetic equations. When an

intra-world distribution function $F$ is associated with a conserved quantity (*e.g.* number of particles or energy), the corresponding hydrodynamic equation is typically expected to display diffusive behavior. Indeed, the timescale associated with the conserved quantity is much slower than the other modes: those can be effectively replaced by their local-equilibrium value in a fixed background of $F$, typically resulting in a diffusive term of the type $\nabla_X^2 F$. Here, in the inter-world case, the distribution function $F_{\alpha\beta}$ is not associated with a conserved quantity (until proven otherwise) and there is no clear separation of timescales in the PDEs (37) governing the dynamics of $\phi$ and $\phi_1$. Consequently, one cannot *a priori* apply the standard hydrodynamic approach, and one must solve for the dynamics $\phi$ and $\phi_1$ on an equal footing.

## 4.3 Validating the ansatz numerically

We perform two independent checks of the ansatz proposed in Eq. (31) by comparing, on the one hand, the solutions of the inter-world kinetic equation in (17) computed with the full-fledged collision integral in Eq. (30) with, on the other hand, the solutions of the coupled PDEs in (37). The first check is performed at early times in a near-critical regime while the second check is performed at later times and at a finite distance from criticality.

Numerically solving the kinetic equation is a formidable task which we simplify as much as possible by working in one dimension, $d = 1$, with a regular lattice dispersion $\epsilon_k = -\cos(k)$ for $k \in [-\pi, \pi)$, and a point-like Fermi surface located at the wave-vector $k_F = \pi/2$. We measure energies in units of the half-bandwidth. Note that superconductivity in $d = 1$ is known to be quite different from dimensions $d \geq 2$, with the RPA treatment that we have set up in Section 3 not suited for $d = 1$. However, the objective here is to put to test the ansatz in conditions that are qualitatively similar to $d \geq 2$, and not to correctly capture the peculiar one-dimensional physics. This is the reason why we can afford to work in $d = 1$. We further simplify the computation by working in a non-self-consistent scheme, with the quasi-particle retarded Green's function reading

$$G^R(\omega, k) = \frac{1}{\omega - \epsilon_k + i\Gamma}, \tag{45}$$

and where the Cooperon Green's function $D^R(\omega, k)$ is computed following Eq. (23). $\Gamma > 0$ sets a bare fermionic inverse lifetime. In practice, $\Gamma$ helps the numerical convergence of our algorithms and we set it as the smallest energy scale in the problem. Finally, we further reduce the difficulty by working with on-shell quantities: $\tilde{F}_{\alpha\beta}(k) := F_{\alpha\beta}(\omega = \epsilon_k, k)$.

### 4.3.1 Early times

The first test consists in numerically solving the kinetic equation (17) with the full-fledged collision integral in Eq. (30) for as long as we can ensure numerical convergence of the solutions. In practice, this is challenging and we can only access early times, *i.e.* on the order of fractions of $\tau_F$. Therefore, in order to benchmark the ansatz in both regimes $\phi \sim 1$ and $\phi \sim 0$, we work with an initial condition that simultaneously spans those two regimes. We choose an initial condition with a perturbation of $\tilde{F}_{\alpha\beta}^{\mathrm{corr}}(k)$ in the shape of a Gaussian droplet of large amplitude $\delta\phi_0 \lesssim 1$ and width $R_0$, and localized around $X = 0$. Explicitly, rescaling time and space according to Eq. (39), we take the following symmetric initial condition

$$\tilde{F}_{\alpha\beta}(\tau = 0, X; k) = [1 - \delta\phi_0(X)]\tilde{F}_{\alpha\beta}^{\mathrm{corr}}(k) \longleftrightarrow \begin{cases} \phi(\tau = 0, X) = 1 - \delta\phi_0(X), \\ \phi_1(\tau = 0, X) = 0, \end{cases} \tag{46a}$$

$$\text{with } \delta\phi_0(X) = \delta\phi_0 \exp\left[-X^2/\left(2R_0^2\right)\right]\Theta(3R_0 - |X|), \tag{46b}$$

and where $\Theta(X)$ is the Heaviside step function. We found such a Gaussian-shaped droplet, defined on the support $[-3R_0, 3R_0]$, to be easier to time-evolve numerically than a semi-circular

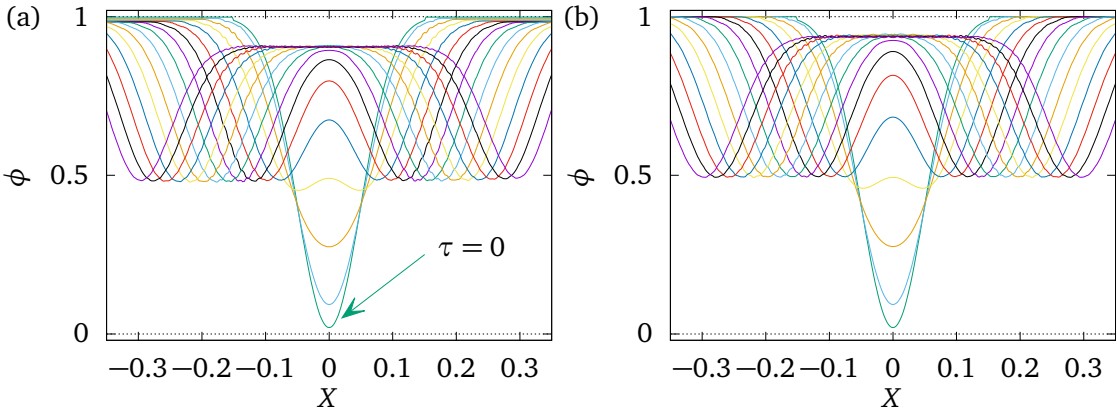

Figure 4: (a) $\phi(\tau, X)$ extracted from the solution $\tilde{F}_{du}(\tau, X; k_F)$ to the full-fledged 1D kinetic equation and plotted at different times $\tau = 0, 0.02, 0.04, \ldots, 0.3$. The initial condition is given in Eq. (46) with $R_0 = 0.05$ and $\delta\phi_0 = 0.98$. (b) Solution $\phi(\tau, X)$ to the coupled PDEs in (37) at $\gamma = 1$ and with the corresponding initial condition. The parameters are $U = -1$, $T = 0.1$, and $\Gamma = 0.01$ (in units of the half-bandwidth), corresponding to a small detuning from criticality $r = -1/(\rho_F D^R(\omega = 0, k = 0)) \approx 0.1$ defined in Eq. (26). Time and space have been rescaled according to Eq. (39). No adjustable parameters were used.

droplet with sharp edges. The physical parameters are chosen such as to be close to criticality (on the disordered side) and to obey the hierarchy $|U|, \epsilon_F \gg T \gg \Gamma$, where $\epsilon_F$ is the Fermi energy.

In Fig. 4, we compare the solutions $\phi(\tau, X)$ of the corresponding coupled PDEs with the solutions $\tilde{F}_{du}(\tau, X; k)$ of the full-fledged kinetic equation. The comparison is made by extracting the first partial-wave contributions according to Eq. (32) which, in 1D, simply reads

$$\begin{cases} \phi(\tau, X) = \frac{1}{2}\left[\tilde{F}_{du}(\tau, X; k_F) + \tilde{F}_{du}(\tau, X; -k_F)\right], \\ \phi_1(\tau, X) = \frac{1}{2}\left[\tilde{F}_{du}(\tau, X; k_F) - \tilde{F}_{du}(\tau, X; -k_F)\right], \end{cases} \tag{47}$$

up to the time $\tau = 0.3$. The qualitative agreement is excellent. Notice the splitting of the initial central perturbation into both a left-moving and a right-moving front. We repeated this analysis in a wide range of near-critical parameters and initial conditions and consistently found excellent agreement, even at a finite distance from criticality, in the presence of fast bosonic fluctuations. This validates the ansatz in Eq. (31) at early times.

### 4.3.2 Late times and partial-wave truncation

Because of the difficulty to produce converged numerical solutions of the kinetic equation at larger times, we resort to a simpler, yet non-trivial, benchmark for the ansatz. Let us consider the case of a spatially homogeneous initial condition but with non-zero anisotropic components in momentum space. Explicitly, we take the initial condition

$$\tilde{F}_{ab}(\tau = 0; k) = \left[\phi_0 + \phi_{10}\text{sign}(k)\right]\tilde{F}_{ab}^{\text{corr}}(k), \tag{48}$$

on the kinetic equation side and, correspondingly,

$$\phi(\tau = 0) = \phi_0 \quad \text{and} \quad \phi_1(\tau = 0) = \phi_{10}, \tag{49}$$

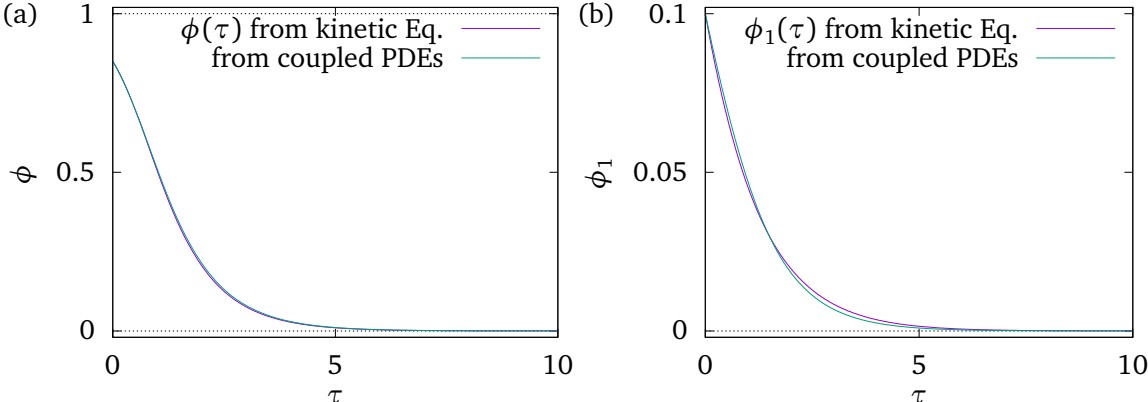

Figure 5: (a) $\phi(\tau)$ extracted from the solution $\tilde{F}_{du}(\tau; k_{\mathrm{F}})$ to the kinetic equation with the spatially homogeneous and momentum anisotropic initial condition given by Eq. (48) with $\phi_0 = 0.85$ and $\phi_{10} = 0.1$, and compared with the solution $\phi(\tau)$ to the coupled PDEs in Eq. (50). (b) Same for the anisotropic component $\phi_1(\tau)$ where $\gamma = 0.5$ is the only adjustable parameter. The physical parameters are $U = -2$, $T = 0.6$, and $\Gamma = 0.2$, corresponding to a sizable detuning from criticality $r = -1/(\rho_{\mathrm{F}} D^R(\omega = 0, k = 0)) \approx 3.9$. Time has been rescaled according to Eq. (39).

on the side of the coupled PDEs which now read

$$\begin{cases} \partial_\tau \phi = \phi(\phi^2 - 1), \\ \partial_\tau \phi_1 = \phi_1(\gamma \phi^2 - 1). \end{cases} \tag{50}$$

Note that these coupled ordinary-differential equations correspond to the discussion around Eq. (42). Notably, Eq. (50) predicts that the relaxation dynamics of $\phi$ (but not those of $\phi_1$) are independent of the distance to criticality, parameterized by $\gamma$. Therefore, complementary to the previous benchmark in Sect. 4.3.1, we test the ansatz at a finite distance from criticality (on the disordered side) by choosing an off-critical set of physical parameters $U$, $T$, and $\Gamma$.

In Fig. 5, we compare the solutions $\phi(\tau)$ and $\phi_1(\tau)$ to the corresponding coupled PDEs with the solutions $\tilde{F}_{du}(\tau, k)$ to the kinetic equation. The comparison is made by extracting the first partial-wave contributions according to Eq. (47). The qualitative agreement is very good from early times down to late times when the dynamics have converged to the uncorrelated world solution. The agreement for the dynamics of $\phi_1(\tau)$ was made by manually adjusting the off-critical value for $\gamma$ given in the caption. This validates the partial-wave truncation which is made in the ansatz.

## 5 Dynamics of information scrambling

In this Section, we solve the set of coupled PDEs (40) that effectively govern the dynamics of information scrambling. We first discuss the early times, when a regime of exponential growth takes place. Later, we solve the geometry for the late-time traveling front. Finally, we address the saturation regime in the bulk of the information light cone.

### 5.1 Early-time exponential growth

At early times, the solutions to the inter-world kinetic equation and to the simplified coupled PDEs are expected to be strongly dependent on the system parameters and the initial conditions.

Here, we solve the coupled PDEs in the linear regime around the correlated-world solution, $\phi \approx 1$ and $\boldsymbol{\phi}_1 \approx 0$. We expect this linear regime to be all the more valid as the initial perturbation will be small, hence taking a longer time to reach the non-linear regime. To that end, we consider the droplet-shaped initial condition

$$\begin{cases} \phi(\tau=0,\boldsymbol{X}) = 1 - \delta\phi_0(\boldsymbol{X}), \\ \boldsymbol{\phi}_1(\tau=0,\boldsymbol{X}) = 0, \end{cases} \tag{51}$$

where the perturbation $0 < \delta\phi_0(\boldsymbol{X}) \ll 1$ is non-vanishing on a small compact ball of radius $R_0$ around $\boldsymbol{X} = 0$. Note that this is a different regime from the numerics presented in Sect. 4.3.1 where the initial condition was probing the non-linear regime. To simplify, we consider the case $d = 1$. The linearized coupled PDEs on $\delta\phi := 1 - \phi$ and $\phi_1$ read

$$\begin{cases} \partial_\tau \delta\phi - \partial_X \phi_1 = 2\delta\phi, \\ \partial_\tau \phi_1 - \partial_X \delta\phi = (\gamma-1)\phi_1. \end{cases} \tag{52}$$

This yields the following linearized PDE on $\delta\phi(\tau,X)$

$$\partial_\tau^2 \delta\phi - (\gamma+1)\partial_\tau \delta\phi - \partial_X^2 \delta\phi = 2(1-\gamma)\delta\phi, \tag{53}$$

with $\delta\phi(\tau=0,X) = \delta\phi_0(X)$ and $\partial_\tau \delta\phi(\tau=0,X) = 2\delta\phi_0(X)$. Integrating the above equation over the whole space, or equivalently the first equation in (52), and introducing the integrated perturbation $\delta M(\tau) := \int dX\, \delta\phi(\tau,X)$, we find an exponential growth of the perturbation, echoing the onset of chaos:

$$\delta M(\tau \ll \tau_*) = \exp\left[2(\tau - \tau_*)\right], \tag{54}$$

with the typical timescale to escape from the unstable solution given by $\tau_* := -\frac{1}{2}\log \delta M_0$ where $\delta M_0 := \int dX\, \delta\phi_0(X)$. The corresponding growth rate, $\lambda_{\mathrm{L}} = 2/\tau_{\mathrm{F}}$ in the original units, can be interpreted as a Lyapunov exponent. Note that the dependence on $\gamma$, the parameter quantifying the distance to criticality, has dropped. A more sophisticated calculation restricted to $1 \ll \tau \ll \tau^*$ and $|X| \ll \tau^{3/4}$ yields the following solution to Eq. (53)

$$\delta\phi(\tau,X) \approx e^{2(\tau-\tau_*)} \frac{e^{-(3-\gamma)X^2/4\tau}}{\sqrt{4\pi\tau/(3-\gamma)}}. \tag{55}$$

See Appendix B for the details of the computation. The above expression involves an exponential growth and a diffusion kernel. It is similar to the solution obtained in the context of the $\mathcal{O}(N)$ model in Ref. [8] (see Eq. 1.13 therein). Let us provide a quick way to justify this solution. We first absorb the exponential growth of $\delta\phi$ that was found in the solution (54) by working in terms of $g(\tau,X) := e^{-2\tau}\delta\phi(\tau,X)$. Using Eq. (53), it obeys

$$\partial_\tau^2 g + (3-\gamma)\partial_\tau g - \partial_X^2 g = 0, \tag{56}$$

with $g(\tau=0,X) = \delta\phi_0(X)$ and $\partial_\tau g(\tau=0,X) = 0$. Next, given that the solution of the above PDE is expected to vary slowly with time, especially at large times, we may neglect the $\partial_\tau^2 g$ term. This yields a simple diffusion equation which, once we switch back to working with $\delta\phi$, is solved by the expression in (55). Note that the solution in (55) can also be seen as the solution to the diffusion+growth equation:

$$\partial_\tau \delta\phi - \frac{1}{3-\gamma}\partial_X^2 \delta\phi = 2\delta\phi, \tag{57}$$

that appears, notably, in the context of branching Brownian motion where it describes the evolution of the expected density of particles [33].

## 5.2 Late-time solutions and discontinuous front

Here, we access the late-time inter-world distribution by analytically solving the coupled PDEs (40) in generic spatial dimensions $d$, simply assuming a spherically-symmetric initial condition. In particular, we shall show that a wavefront propagates at a constant butterfly velocity controlled by the Fermi velocity. The wavefront acts as a light cone that separates two causally disjoint regions: ahead of the front, the inter-world distribution is the correlated-world solution, while behind the front the two worlds rapidly decohere to the uncorrelated-world solution. Notably, at the front, the distribution function develops a discontinuity.

### 5.2.1 Traveling front

The solution develops a traveling front located on a sphere of increasing radius. Our goal is to compute its velocity and shape.

The first step is to notice that the late-time position of the front is, by definition, far from the origin and we may neglect the $1/r$ term in the RHS of Eq. (41). By doing so, we simply recover the $d = 1$ equations. Indeed, given the large radius of the sphere where the front is located, the problem is locally flat in the non-radial directions. Therefore, we only need to work out the $d = 1$ case. To simplify the presentation of the computation, we work out the critical case where $\gamma = 1$. However, the generic case for $\gamma \neq 1$ can also be solved by similar techniques and we refer the reader to Appendix C for the corresponding detailed computation. Introducing the fields

$$\varphi := \frac{\phi + \phi_1}{2} \quad \text{and} \quad \psi := \phi - \phi_1, \tag{58}$$

the coupled PDEs can be cast as

$$\begin{cases} \partial_\tau \varphi + \partial_X \varphi = \varphi \left( \phi^2 - 1 \right), \\ \partial_\tau \psi - \partial_X \psi = \psi \left( \phi^2 - 1 \right), \end{cases} \tag{59}$$

with the initial conditions,

$$\begin{cases} \varphi(\tau = 0, X) = \phi_0(X)/2, \\ \psi(\tau = 0, X) = \phi_0(X). \end{cases} \tag{60}$$

The method of characteristics gives the implicit solutions

$$\begin{cases} \varphi(\tau, X) = \frac{1}{2} \phi_0(X - \tau) e^{-\int_0^\tau ds \left[ 1 - \phi^2(s, X - \tau + s) \right]}, \\ \psi(\tau, X) = \phi_0(X + \tau) e^{-\int_0^\tau ds \left[ 1 - \phi^2(s, X + \tau - s) \right]}, \end{cases} \tag{61}$$

which yield

$$\phi(\tau, X) = \frac{1}{2} \phi_0(X - \tau) e^{-\int_0^\tau ds \left[ 1 - \phi^2(s, X - \tau + s) \right]} + \frac{1}{2} \phi_0(X + \tau) e^{-\int_0^\tau ds \left[ 1 - \phi^2(s, X + \tau - s) \right]}. \tag{62}$$

Let us now assume that a right-moving front, traveling at velocity 1 (in units of $v_F / \sqrt{d}$), develops at late times, *i.e.*

$$\phi_+(\tau, X) := \phi(\tau, X + \tau) \overset{\tau \to \infty}{\to} f_+(X), \tag{63}$$

pointwise. Naturally, there is also a symmetrical left-moving front. We start from

$$\phi_+(\tau, X) = \frac{1}{2} \phi_0(X) e^{-\int_0^\tau ds \left[ 1 - \phi_+^2(s, X) \right]} + \frac{1}{2} \phi_0(X + 2\tau) e^{-\int_0^\tau ds \left[ 1 - \phi_+^2(s, X + 2\tau - 2s) \right]} \tag{64}$$

$$= \frac{1}{2} \phi_0(X) e^{-\int_0^\tau ds \left[ 1 - \phi_+^2(s, X) \right]} + \frac{1}{2} \phi_0(X + 2\tau) e^{-\int_0^\tau ds \left[ 1 - \phi_+^2(\tau - s, X + 2s) \right]}, \tag{65}$$

where in the second line we performed a change of dummy variable $s \to \tau - s$. In the long-time limit $\tau \to \infty$, we get

$$f_+(X) = \frac{1}{2}\phi_0(X)e^{-\int_0^\infty ds\left[1-\phi_+^2(s,X)\right]} + \frac{1}{2}e^{-\int_0^\infty ds\left[1-f_+^2(X+2s)\right]}, \qquad (66)$$

where we used the property $\phi_0(X + 2\tau) \to 1$. Consistently with the Eq. (62), we postulate[1] that the front is such that

$$f_+(X < R_0) < 1 \quad \text{and} \quad f_+(X > R_0) = 1. \qquad (67)$$

Let us now work with $X < R_0$. Hence, $\lim_{s\to\infty} 1 - \phi_+^2(s, X < R_0) = 1 - f_+^2(X) > 0$, which implies that the first term in Eq. (66) is zero. After the change of variables $X + 2s \to u$, we are left with

$$f_+(X < R_0) = \frac{1}{2}e^{-\frac{1}{2}\int_X^{R_0} du\left[1-f_+^2(u)\right]}. \qquad (68)$$

At $X \xrightarrow{X<R_0} R_0$, this yields $f_+(R_0) = 1/2$. Given that we assume $f_+(X > R_0) = 1$, this signals a discontinuity in $f_+(X)$ at $X = R_0$. Moreover, Eq. (68) implies that $f_+(X)$ obeys

$$f_+'(X) = \frac{1}{2}\left[1 - f_+^2(X)\right]f_+(X), \qquad (69)$$

which can be solved by separation of variables, yielding the discontinuous right-moving front with the shape

$$f_+(X < R_0) = \frac{1}{\sqrt{1 + 3e^{R_0-X}}} \xrightarrow{X\to R_0^-} \frac{1}{2} \quad \text{and} \quad f_+(X > R_0) = 1. \qquad (70)$$

Note that, here, the information on the precise shape of the initial perturbation is lost. Except for a trivial spatial offset in the shape of the front, $R_0$ drops out of the problem. This is expected for generic initial conditions defined on a compact support $|X| < R_0$. Indeed, one can show that the first non-vanishing derivative $\partial_X^{(n)}\phi_0(X = R_0^-) > 0$ is responsible for the generation of a first-order derivative $f_+(X = R_0^-) > 0$ which grows exponentially with time, therefore creating a discontinuity at large times. This independence of the steady-state solution with respect to the initial condition is the hallmark of a universal solution. Notably, as previously discussed in Sec. 5.1, $R_0$ and the precise shape of the initial perturbation are however controlling the timescale for the traveling front to form. Barring this point, the universal features of the steady state may be safely accessed by sending $R_0 \to 0^+$ after $\tau \to \infty$. We obtain the right-moving traveling front

$$\begin{cases} \lim_{\tau\to\infty} \phi(\tau, X + \tau) = f_+(X), \\ \lim_{\tau\to\infty} \phi_1(\tau, X + \tau) = f_{1+}(X), \end{cases} \qquad (71)$$

with

$$\begin{cases} f_+(X < 0) = \frac{1}{\sqrt{1+3e^{-X}}} \xrightarrow{X\to 0^-} \frac{1}{2} \quad \text{and} \quad f_+(X > 0) = 1, \\ f_{1+}(X < 0) = -\frac{1}{\sqrt{1+3e^{-X}}} \xrightarrow{X\to 0^-} -\frac{1}{2} \quad \text{and} \quad f_{1+}(X > 0) = 0. \end{cases} \qquad (72)$$

The last equation above follows from $\phi_1 = \varphi - \psi/2$. As we discussed above, the velocity and the precise shape of the late-time traveling front, $f_+(X)$, and notably its discontinuity,

---

[1]This assumption can be rigorously proven to be true in $d = 1$.

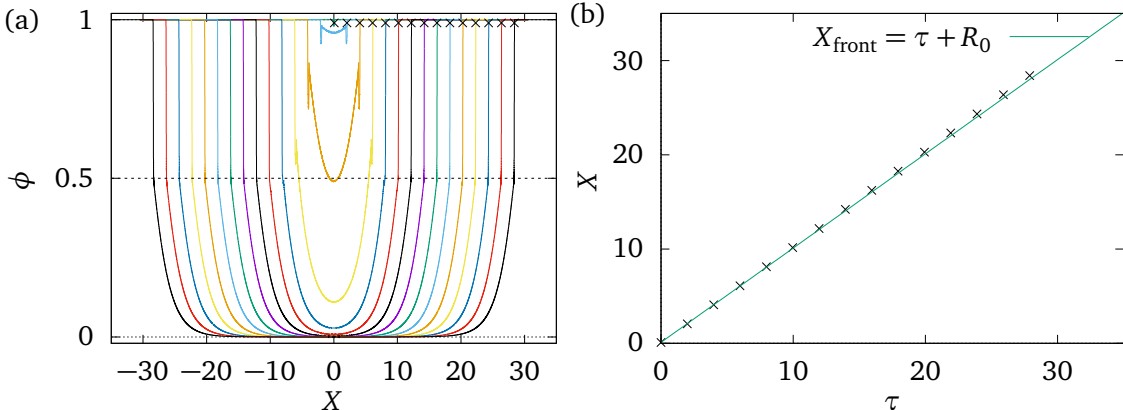

Figure 6: (a) Solution $\phi(\tau, X)$ to the $d = 1$ coupled PDEs in Eq. (40) at $\gamma = 1$ for the droplet initial perturbation in Eq. (75) with amplitude $\delta\phi_0 = 0.02$ and radius $R_0 = 0.1$. Different times $\tau = 0, 2, \ldots, 26, 28$ are plotted with different colors. (b) Location of the front extracted from the data in (a) marked by crosses. The line is the analytical prediction, with a constant butterfly velocity, $X_{\text{front}}(\tau) = \tau + R_0$. No adjustable parameter was used.

generalize to the radial front $f_+(r)$ in generic dimension $d$ (assuming spherically-symmetric initial conditions). In terms of the original inter-world distribution function, reinstating the original units of time and space, the traveling front reads

$$\lim_{t \to \infty} \tilde{F}_{du}(\boldsymbol{k}; t, r + v_{\text{B}}t)\big|_{k \to k_{\text{F}}} = \begin{cases} f_+\left(\frac{r\sqrt{d}}{\ell}\right)[1 - \boldsymbol{u_k} \cdot \boldsymbol{u_r}] & \text{if } r < 0, \\ 1 & \text{if } r > 0, \end{cases} \tag{73}$$

with the butterfly velocity given by $v_{\text{B}} := v_{\text{F}}/\sqrt{d}$ and the radial front shape $f_+$ given by Eq. (72) that spans over a length scale controlled by the mean free path $\ell := v_{\text{F}}\tau_{\text{F}}$. We recall that $v_{\text{F}}$ is the Fermi velocity and the scattering time $\tau_{\text{F}}$ was defined in Eq. (38).

This solution can also be generalized to cases away from criticality, *i.e.* for a generic $\gamma \neq 1$. The qualitative features are found to be very similar to the critical case $\gamma = 1$. In the Appendix C, we show that the discontinuity of the front is now from $f_+(X = 0^-) = L(\gamma)$ to $f_+(X > 0) = 1$ with

$$L(\gamma) = \frac{\sqrt{5 + 4\gamma} - 1}{2(1 + \gamma)}. \tag{74}$$

This quantity monotonically interpolates from $L = 1/2$ for $\gamma = 1$ to $L = 1/(\text{golden ratio})$ for $\gamma = 0$.

The discontinuous traveling wavefronts in Eqs. (72), (73), and their generalization to non-critical regimes in Eq. (74) are one of the main results of this manuscript. To illustrate the analytical solution, we compare its predictions to the numerical solution of the coupled PDEs at $\gamma = 1$ (critical regime). In Fig. 6, we display the numerical solution starting from the following droplet-shaped initial perturbation of the correlated-world solution

$$\begin{cases} \phi(\tau = 0, X) = 1 - \delta\phi_0(X) & \text{with} \quad \delta\phi_0(X) = \delta\phi_0\sqrt{1 - (X/R_0)^2}\,\Theta(R_0 - |X|), \\ \phi_1(\tau = 0, X) = 0, \end{cases} \tag{75}$$

where $\delta\phi_0$ sets the amplitude of the droplet, and $R_0$ sets its radius. This illustrates unambiguously the spatial growth of the loss of quantum coherence as time goes on. The light cone

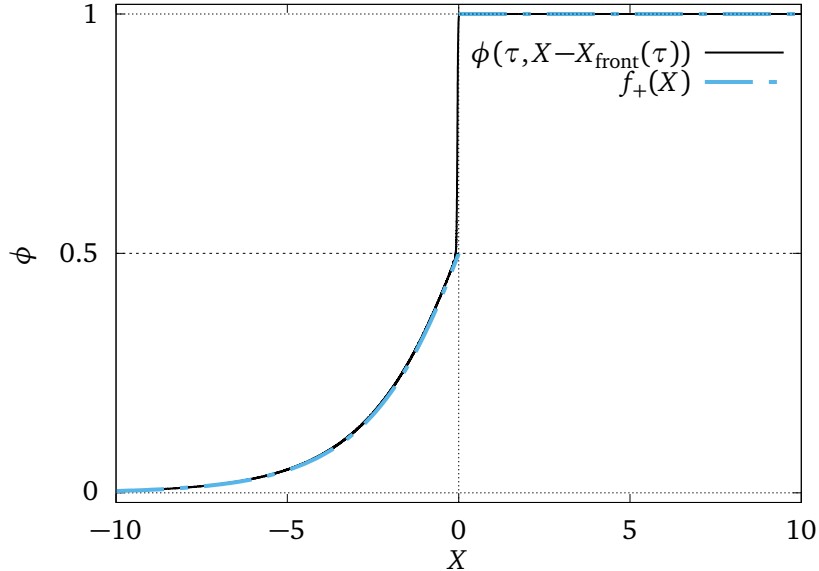

Figure 7: Discontinuous late-time front: the solution $\phi(\tau, X - X_{\text{front}}(\tau))$ is extracted from the data of Fig. 6 (a) for $\tau = 28$ (solid line), and compared to the analytical prediction $f_+(X)$ t in Eq. (72) for the traveling front, with a discontinuity at $X = 0$ from 0.5 to 1 (dashed line). No adjustable parameter was used.

structure of this growth, with a front traveling at a constant velocity, is demonstrated by simply extracting the location of the front as a function of time. In Fig. 7, we illustrate the discontinuous shape of the traveling front by superimposing the front extracted from the numerical solution of $\phi(\tau, X)$ at the late-time $\tau = 28$ to the exact result given by $f_+(X)$ in Eq. (72).

### 5.2.2 Saturation inside the light cone

Inside the light cone and far enough from its boundaries, we have $\phi \ll 1$ and we can neglect the non-linearities in the RHS of the coupled PDEs (59). Working in the $d = 1$ case, the linearized PDEs read

$$\begin{cases} \partial_\tau \phi + \partial_X \phi_1 = -\phi\,, \\ \partial_\tau \phi_1 + \partial_X \phi = -\phi_1\,. \end{cases} \tag{76}$$

Assuming symmetric initial conditions, *i.e.*,

$$\phi(\tau = 0, X) = \phi(\tau = 0, -X) \quad \text{and} \quad \phi_1(\tau = 0, X) = -\phi_1(\tau = 0, -X)\,,$$

one can simply show that this symmetry is preserved by the entire time evolution (even in the presence of the non-linear terms that were at play in the earlier regime). The solutions are of the form

$$\begin{cases} \phi(\tau, X) = e^{-X} f_\varphi(X - \tau) + e^X f_\varphi(-X - \tau)\,, \\ \phi_1(\tau, X) = e^{-X} f_\varphi(X - \tau) - e^X f_\varphi(-X - \tau)\,, \end{cases} \tag{77}$$

where the function $f_\varphi(X)$ should in principle be determined by solving the early-to-intermediate time problem. In practice, we can determine the asymptotic behavior of the function $f_\varphi(X)$ at large $X$ by requiring a matching to the left side of the late-time front computed previously:

$$\lim_{\tau \to \infty} \phi(\tau, X + \tau) \sim f_+(X), \quad \text{for} \quad X < 0, \ |X| \gg 1, \tag{78}$$

where $f_+(X)$ was computed in Eq. (72). Using the late-time solution in Eq. (77), we have

$$\lim_{\tau \to \infty} e^{-X-\tau} f_\varphi(X) + e^{X+\tau} f_\varphi(-X - 2\tau) \sim f_+(X). \tag{79}$$

The first term vanishes and we are left with

$$\lim_{\tau \to \infty} e^{X+\tau} f_\varphi(-X - 2\tau) \sim f_+(X), \tag{80}$$

which is solved as

$$f_\varphi(X) \sim e^{X/2}. \tag{81}$$

This yields, at large times and far inside the boundary of the light cone, *i.e.* for $\tau - |X| \gg 1$,

$$\begin{cases} \phi(\tau, X) \simeq 2e^{-\tau/2} \cosh(X/2), \\ \phi_1(\tau, X) \simeq -2e^{-\tau/2} \sinh(X/2). \end{cases} \tag{82}$$

These solutions can also be obtained directly from injecting the expression (72) for the traveling front in Eq. (61), using the spatial symmetry $\lim_{\tau \to \infty} \phi(\tau, \tau + X) = \lim_{\tau \to \infty} \phi(\tau, -\tau - X) = f_+(X)$. Note that these asymptotic solutions have already lost the information about the initial condition. Moreover, they imply the relation $\phi_1(\tau, X) = -2\partial_X \phi(\tau, X)$ for $\tau - |X| \gg 1$. Once re-injected in the linearized PDEs, this yields the following linearly-driven diffusion equation for $\phi(\tau, X)$ only

$$\partial_\tau \phi - 2\partial_X^2 \phi = -\phi, \tag{83}$$

where the diffusive constant in front of the $\partial_X^2 \phi$ term is $D := 2v_F^2 \tau_F$ once the original units of time and space are reinstated. We note that the diffusive LHS of Eq. (83) is similar to the one of the FKPP-like equation that was derived in a similar context in Ref. [15], see Eq. (76) therein.

## 6 Discussion and conclusion

Starting from the microscopic Hamiltonian of a $d$-dimensional quantum many-body system of interacting electrons close to a superconducting phase transition, we carefully derived the corresponding dynamics of quantum information scrambling.

Quite expectedly, we found a ballistic spread of information governed by a non-universal butterfly velocity $v_B$. We presented analytical solutions in the different regimes relevant to quantum chaotic dynamics: the early exponential growth, the geometry of the late-time front, and the saturation within the light cone.

Perhaps the most striking result of our work is the fact that the scrambling of quantum information at late times is governed by shock-wave dynamics. Scrambling propagates at the maximum velocity allowed by causality and develops a distinct discontinuity exactly at the boundary of the light cone. Notably, these dynamics scrupulously respect causality: scrambling does not leak outside of the light cone. This is different from the findings of previous works in similar settings, which often exhibited sharp but continuous fronts preceded by exponential tails. At a formal level, this difference arises from the fact that our effective dynamics are governed by a set of coupled PDEs that does not belong to the reaction-diffusion class.

While our full solution does share strong similarities with the diffusive FKPP solutions in the linearized regimes (at early times or deep in the bulk of the light cone), we attribute this explicit absence of a diffusive term to the fact that information dynamics is not directly associated with a conserved quantity, unlike usual transport which is associated with, say, charge, energy, or momentum conservation.

Addressing the robustness of the finite spacetime discontinuity in the traveling front is perhaps one of the most pressing questions raised by our results. Let us first note that the discontinuity has to be understood on the scale of the mean free path set by the inelastic scattering, $\ell := v_F \tau_F$, and not on the microscopic scale $1/k_F$ where the kinetic equation breaks down. The discontinuity is unlikely an artifact of the one-loop RPA scheme (exact in the limit where the number of electronic orbitals is sent to infinity), nor of our truncated partial-wave expansion in momentum space (key to the ensuing strict causal structure, and likely to be a very good approximation in some geometries). However, higher-order terms in the Moyal product expansion that were neglected in the quasi-classical approximation, or any source of noise or disorder [34], *e.g.* caused by elastic scattering on random impurities, could smoothen the front at the light-cone boundary and possibly decrease the butterfly velocity. Numerical confirmation of the shock-wave dynamics starting from a microscopic Hamiltonian is expected to be difficult as the scrambling front propagates fast and the discontinuity only develops at late times. This means that one should simulate large systems of linear size $L \sim 10\,\ell$ until long times $t_{max} \sim 10\,\tau_F$. Tensor network approaches in 1d may be suited to meet the challenge [4].

# Acknowledgements

The authors are grateful to Adam Nahum and Tianci Zhou for insightful discussions. AM thanks the hospitality of École Normale Supérieure (Paris) where this work was initiated.

**Funding information** CA acknowledges the support from the French ANR "MoMA" project ANR-19-CE30-0020 and the support from the project 6004-1 of the Indo-French Centre for the Promotion of Advanced Research (IFCPAR). This work received support from the U.S. National Science Foundation Grant No. NSF DMR-2018358 (A.M.). AM acknowledges the Aspen Center for Physics where part of this work was performed and which is supported by the US National Science Foundation Grant PHY-1607611.

# A  Inter-world collision integral

In this Appendix, we carefully derive the contributions of the interworld collision integral to the coupled PDEs (37). We recall the collision integral, onshell:

$$\tilde{I}_{\alpha\beta}(\boldsymbol{k}) = 2\,\mathrm{Im}\,\tilde{\Sigma}^R(\boldsymbol{k})\,\tilde{F}_{\alpha\beta}(\boldsymbol{k}) + i\tilde{\Sigma}^K_{\alpha\beta}(\boldsymbol{k}), \tag{A.1}$$

where

$$\begin{aligned}
2\,\mathrm{Im}\,\tilde{\Sigma}^R(\boldsymbol{k}) = 2\sum_{\boldsymbol{k}'\boldsymbol{k}''}\int &\frac{d\omega'}{2\pi}\frac{d\omega''}{2\pi}|D^R(\omega',\boldsymbol{k}')|^2 \\
&\times \mathrm{Im}\,G^R(\omega'',\boldsymbol{k}'')\,\mathrm{Im}\,G^R(\omega'-\omega'',\boldsymbol{k}'-\boldsymbol{k}'')\,\mathrm{Im}\,G^R(\omega'-\epsilon_{\boldsymbol{k}},\boldsymbol{k}'-\boldsymbol{k}) \\
&\times \left[\tanh\left(\frac{\omega''}{2T}\right) + \tanh\left(\frac{\omega'-\omega''}{2T}\right)\right]\left[\coth\left(\frac{\omega'}{2T}\right) + \tanh\left(\frac{\epsilon_{\boldsymbol{k}}-\omega'}{2T}\right)\right],
\end{aligned} \tag{A.2}$$

and

$$
\begin{aligned}
i\tilde{\Sigma}^K_{\alpha\beta}(\boldsymbol{k}) = 2\sum_{\boldsymbol{k}'\boldsymbol{k}''}\int \frac{d\omega'}{2\pi}\frac{d\omega''}{2\pi}|D^R(\omega',\boldsymbol{k}')|^2\,\text{Im}\,G^R(\omega'',\boldsymbol{k}'')\,\text{Im}\,G^R\left(\omega'-\omega'',\boldsymbol{k}'-\boldsymbol{k}''\right) \\
\times\,\text{Im}\,G^R\left(\omega'-\epsilon_{\boldsymbol{k}},\boldsymbol{k}'-\boldsymbol{k}\right)\tilde{F}_{\alpha\beta}(\boldsymbol{k}'')\tilde{F}_{\alpha\beta}(\boldsymbol{k}'-\boldsymbol{k}'')\tilde{F}_{\beta\alpha}(\boldsymbol{k}'-\boldsymbol{k}).
\end{aligned}
\tag{A.3}
$$

Now let us inject the following ansatz in Eq. (A.1),

$$
\tilde{F}^{\text{ansatz}}_{\alpha\beta}(\boldsymbol{k}) = \left[\phi + \boldsymbol{u}_{\boldsymbol{k}}\cdot\boldsymbol{\phi}_1\right]\tilde{F}^{\text{corr}}_{\alpha\beta}(\boldsymbol{k}).
\tag{A.4}
$$

The term in $\text{Im}\tilde{\Sigma}^R(\boldsymbol{k})$ is straight-forward:

$$
2\,\text{Im}\tilde{\Sigma}^R(\boldsymbol{k})\left[\phi + \boldsymbol{u}_{\boldsymbol{k}}\cdot\boldsymbol{\phi}_1\right]\tilde{F}^{\text{corr}}_{\alpha\beta}(\boldsymbol{k}).
\tag{A.5}
$$

Let us treat the term $i\tilde{\Sigma}^K_{\alpha\beta}(\boldsymbol{k})$ in Eq. (A.3). It produces terms in $\phi^3$, $\phi^2\phi_1$, $\phi\phi_1^2$, and $\phi_1^3$. In practice, consistently with our choice of ansatz which consists of tracking only the two first multipolar contributions to $\tilde{F}(\boldsymbol{k})$, we discard the terms of order $\phi_1^2$ and $\phi_1^3$ which yield higher-order multipolar contributions to the collision integral. The term in $\phi^3$ reads

$$
\begin{aligned}
2\phi^3\sum_{\boldsymbol{k}'\boldsymbol{k}''}\int \frac{d\omega'}{2\pi}\frac{d\omega''}{2\pi}|D^R(\omega',\boldsymbol{k}')|^2\,\text{Im}\,G^R(\omega'',\boldsymbol{k}'')\,\text{Im}\,G^R(\omega'-\omega'',\boldsymbol{k}'-\boldsymbol{k}'') \\
\times\,\text{Im}\,G^R(\omega'-\epsilon_{\boldsymbol{k}},\boldsymbol{k}'-\boldsymbol{k})\tilde{F}^{\text{corr}}_{\alpha\beta}(\boldsymbol{k}'')\tilde{F}^{\text{corr}}_{\alpha\beta}(\boldsymbol{k}'-\boldsymbol{k}'')\tilde{F}^{\text{corr}}_{\beta\alpha}(\boldsymbol{k}'-\boldsymbol{k}) \\
= 2\phi^3\,\text{Im}\tilde{\Sigma}^R(\boldsymbol{k}),
\end{aligned}
\tag{A.6}
$$

where we used the trigonometric relation

$$
\begin{aligned}
F^{\text{corr}}_{\alpha\beta}(\omega'')F^{\text{corr}}_{\alpha\beta}(\omega'-\omega'')F^{\text{corr}}_{\beta\alpha}(\omega'-\omega) \\
+ \underbrace{\left[\tanh\left(\frac{\omega''}{2T}\right) + \tanh\left(\frac{\omega'-\omega''}{2T}\right)\right]\left[\coth\left(\frac{\omega'}{2T}\right) + \tanh\left(\frac{\omega-\omega'}{2T}\right)\right]}_{\geq 0}F^{\text{corr}}_{\alpha\beta}(\omega) = 0.
\end{aligned}
\tag{A.7}
$$

Let us now evaluate the term of order $\phi^2\phi_1$. We have

$$
\begin{aligned}
i\tilde{\Sigma}^K_{\alpha\beta}(\boldsymbol{k}) = -2\phi^3\,\text{Im}\tilde{\Sigma}^R(\boldsymbol{k})F^{\text{corr}}_{\alpha\beta}(\boldsymbol{k}) \\
+ 2\phi^2\sum_{\boldsymbol{k}'\boldsymbol{k}''}\int \frac{d\omega'}{2\pi}\frac{d\omega''}{2\pi}|D^R(\omega',\boldsymbol{k}')|^2\,\text{Im}\,G^R(\omega'',\boldsymbol{k}'')\,\text{Im}\,G^R(\omega'-\omega'',\boldsymbol{k}'-\boldsymbol{k}'') \\
\times\,\text{Im}\,G^R(\omega'-\epsilon_{\boldsymbol{k}},\boldsymbol{k}'-\boldsymbol{k})\left\{\boldsymbol{\phi}_1\cdot\left(\boldsymbol{u}_{\boldsymbol{k}''} + \boldsymbol{u}_{\boldsymbol{k}'-\boldsymbol{k}''} + \boldsymbol{u}_{\boldsymbol{k}'-\boldsymbol{k}}\right)\right\}F^{\text{corr}}_{\alpha\beta}(\omega'')F^{\text{corr}}_{\alpha\beta}(\omega'-\omega'')F^{\text{corr}}_{\beta\alpha}(\omega'-\omega).
\end{aligned}
\tag{A.8}
$$

The expression can be simplified by use of Eq. (A.7), yielding

$$
\begin{aligned}
i\tilde{\Sigma}^K_{\alpha\beta}(\boldsymbol{k}) = -2\phi^3\,\text{Im}\tilde{\Sigma}^R(\boldsymbol{k})\tilde{F}^{\text{corr}}_{\alpha\beta}(\boldsymbol{k}) \\
- 2\phi^2\tilde{F}^{\text{corr}}_{\alpha\beta}(\boldsymbol{k})\sum_{\boldsymbol{k}'\boldsymbol{k}''}\int \frac{d\omega'}{2\pi}\frac{d\omega''}{2\pi}|D^R(\omega',\boldsymbol{k}')|^2\,\text{Im}\,G^R(\omega'',\boldsymbol{k}'')\,\text{Im}\,G^R(\omega'-\omega'',\boldsymbol{k}'-\boldsymbol{k}'') \\
\times\,\text{Im}\,G^R(\omega'-\epsilon_{\boldsymbol{k}},\boldsymbol{k}'-\boldsymbol{k})\left[\tanh\left(\frac{\omega''}{2T}\right) + \tanh\left(\frac{\omega'-\omega''}{2T}\right)\right]\left[\coth\left(\frac{\omega'}{2T}\right) + \tanh\left(\frac{\epsilon_{\boldsymbol{k}}-\omega'}{2T}\right)\right] \\
\times\left\{\boldsymbol{\phi}_1\cdot(\boldsymbol{u}_{\boldsymbol{k}''} + \boldsymbol{u}_{\boldsymbol{k}'-\boldsymbol{k}''} + \boldsymbol{u}_{\boldsymbol{k}'-\boldsymbol{k}})\right\}.
\end{aligned}
\tag{A.9}
$$

## A.1 Critical case

Close to criticality, the Cooperon propagator reads

$$D^R(\omega, \mathbf{k}) \approx \frac{-1/\rho_{\mathrm{F}}}{r - \mathrm{i}a\omega/T + \xi^2 k^2 + \dots}, \tag{A.10}$$

with the positive parameters $r \propto (T - T_{\mathrm{c}})/T_{\mathrm{c}}$, $a \sim \mathcal{O}(1)$, $\xi^2 \sim v_{\mathrm{F}}^2/T^2$. At criticality $r \to 0$, the Cooperon becomes soft with a diverging length scale $l \sim 1/r^\nu$ (here $\nu = 1/2$), and the propagator is singular at $\omega = k = 0$. In this case, we can approximate the term $\mathbf{u}_{k''} + \mathbf{u}_{k'-k''} + \mathbf{u}_{k'-k} \approx -\mathbf{u}_k$. Thus we have

$$\mathrm{i}\tilde{\Sigma}_{\alpha\beta}^K(\mathbf{k}) = -2\phi^3 \, \mathrm{Im}\tilde{\Sigma}^R(\mathbf{k}) F_{\alpha\beta}^{\mathrm{corr}}(\mathbf{k}) \tag{A.11}$$

$$+ 2\phi^2(\boldsymbol{\phi}_1 \cdot \mathbf{u}_k)\tilde{F}_{\alpha\beta}^{\mathrm{corr}}(\mathbf{k}) \sum_{\mathbf{k}' \mathbf{k}''} \int \frac{\mathrm{d}\omega'}{2\pi} \frac{\mathrm{d}\omega''}{2\pi} |D^R(\omega', \mathbf{k}')|^2 \, \mathrm{Im}\, G^R\left(\omega'', \mathbf{k}''\right) \, \mathrm{Im}\, G^R\left(\omega' - \omega'', \mathbf{k}' - \mathbf{k}''\right)$$

$$\times \mathrm{Im}\, G^R(\omega' - \epsilon_k, \mathbf{k}' - \mathbf{k}) \left[\tanh\left(\frac{\omega''}{2T}\right) + \tanh\left(\frac{\omega' - \omega''}{2T}\right)\right]\left[\coth\left(\frac{\omega'}{2T}\right) + \tanh\left(\frac{\epsilon_k - \omega'}{2T}\right)\right].$$

Performing the sums on $\omega''$ and $\mathbf{k}''$, we get

$$\mathrm{i}\tilde{\Sigma}_{\alpha\beta}^K(\mathbf{k}) = -2\phi^3 \, \mathrm{Im}\tilde{\Sigma}^R(\mathbf{k}) F_{\alpha\beta}^{\mathrm{corr}}(\mathbf{k})$$

$$+ 4\phi^2(\boldsymbol{\phi}_1 \cdot \mathbf{u}_k) \sum_{\mathbf{k}'} \int \frac{\mathrm{d}\omega'}{2\pi} |D^R(\omega', \mathbf{k}')|^2 \mathrm{Im}\Pi^R(\omega', \mathbf{k}')\left[\coth\left(\frac{\omega'}{2T}\right) + \tanh\left(\frac{\epsilon_k - \omega'}{2T}\right)\right]$$

$$\times \mathrm{Im}\, G^R(\omega' - \epsilon_k, \mathbf{k}' - \mathbf{k})\tilde{F}_{\alpha\beta}^{\mathrm{corr}}(\mathbf{k}). \tag{A.12}$$

The above can now be written as

$$\mathrm{i}\tilde{\Sigma}_{\alpha\beta}^K(\mathbf{k}) = -2\phi^2\left[\phi + (\boldsymbol{\phi}_1 \cdot \mathbf{u}_k)\right]\tilde{F}_{\alpha\beta}^{\mathrm{corr}}(\mathbf{k})\mathrm{Im}\tilde{\Sigma}^R(\mathbf{k}). \tag{A.13}$$

Altogether, we obtain

$$\tilde{I}_{\alpha\beta}(\mathbf{k}) = 2(1 - \phi^2)\underbrace{\left[\phi + (\boldsymbol{\phi}_1 \cdot \mathbf{u}_k)\right]\tilde{F}_{\alpha\beta}^{\mathrm{corr}}(\mathbf{k})}_{\tilde{F}_{\alpha\beta}^{\mathrm{ansatz}}(\mathbf{k})} \mathrm{Im}\tilde{\Sigma}^R(\mathbf{k}). \tag{A.14}$$

## A.2 Away from criticality

In the Subsection A.1 above, we have treated the critical case which yields the following coupled PDEs

$$\begin{cases} \partial_t \phi + \frac{v_{\mathrm{F}}}{d}\boldsymbol{\nabla}_x \cdot \boldsymbol{\phi}_1 = \phi(\phi^2 - 1)/\tau_{\mathrm{F}}, \\ \partial_t \boldsymbol{\phi}_1 + v_{\mathrm{F}}\boldsymbol{\nabla}_x \phi = \boldsymbol{\phi}_1(\phi^2 - 1)/\tau_{\mathrm{F}}, \end{cases} \tag{A.15}$$

where $v_{\mathrm{F}}$ is the Fermi velocity and we defined the timescale $\tau_{\mathrm{F}}$ as $1/\tau_{\mathrm{F}} := -2\,\mathrm{Im}\,\tilde{\Sigma}^R(k_{\mathrm{F}})$.

Away from criticality, assuming $1/\tau_{\mathrm{F}} > 0$, we separate the expression of $\mathrm{i}\tilde{\Sigma}_{\alpha\beta}^K(\mathbf{k})$ in Eq. (A.9) into the critical expression computed in Eq. (A.13) and the rest by simply writing

$$\mathbf{u}_{k''} + \mathbf{u}_{k'-k''} + \mathbf{u}_k = -\mathbf{u}_k + \left(\mathbf{u}_{k''} + \mathbf{u}_{k'-k''} + \mathbf{u}_k + \mathbf{u}_{k'-k}\right). \tag{A.16}$$

Explicitly, we have

$$
\mathrm{i}\tilde{\Sigma}^K_{\alpha\beta}(\boldsymbol{k}) = -2\phi^2\big[\phi + (\boldsymbol{\phi}_1\cdot\boldsymbol{u}_k)\big]\tilde{F}^{\mathrm{corr}}_{\alpha\beta}(\boldsymbol{k})\mathrm{Im}\tilde{\Sigma}^R(\boldsymbol{k})
$$

$$
-2\phi^2\tilde{F}^{\mathrm{corr}}_{\alpha\beta}(\boldsymbol{k})\sum_{\boldsymbol{k}'\boldsymbol{k}''}\int\frac{\mathrm{d}\omega'}{2\pi}\frac{\mathrm{d}\omega''}{2\pi}|D^R(\omega',\boldsymbol{k}')|^2\,\mathrm{Im}\,G^R\big(\omega'',\boldsymbol{k}''\big)\,\mathrm{Im}\,G^R\big(\omega'-\omega'',\boldsymbol{k}'-\boldsymbol{k}''\big)
$$

$$
\times\mathrm{Im}\,G^R(\omega'-\epsilon_k,\boldsymbol{k}'-\boldsymbol{k})\underbrace{\left[\tanh\left(\frac{\omega''}{2T}\right)+\tanh\left(\frac{\omega'-\omega''}{2T}\right)\right]\left[\coth\left(\frac{\omega'}{2T}\right)+\tanh\left(\frac{\epsilon_k-\omega'}{2T}\right)\right]}_{\geq 0}
$$

$$
\times\left\{\boldsymbol{\phi}_1\cdot\big(\boldsymbol{u}_{k''}+\boldsymbol{u}_{k'-k''}+\boldsymbol{u}_k+\boldsymbol{u}_{k'-k}\big)\right\}, \tag{A.17}
$$

where the first term is the critical expression while the second term collects the rest. Assuming that $\epsilon_k = \epsilon_{-k}$, one may check that this second term is odd under $\boldsymbol{k}\to-\boldsymbol{k}$ and therefore cannot contribute to the projection on the momentum-space isotropic contribution

$$
\int\mathrm{d}\Omega_k\text{ last term of Eq. (A.17)} = 0\,. \tag{A.18}
$$

This guarantees that the RHS of the first equation in the coupled PDE in (A.15) is also valid away from criticality. However, in the absence of a similar symmetry argument, we expect the projection to the first partial wave to be non-vanishing, *i.e.*

$$
\int\mathrm{d}\Omega_k\boldsymbol{u}_k\text{ last term of Eq. (A.17)}\neq 0\,, \tag{A.19}
$$

and therefore to give an extra contribution to the term in $\phi^2\boldsymbol{\phi}_1$ in the RHS of the second equation in (A.15). Now working at the Fermi surface, we can parameterize, without loss of generality, the amplitude of this contribution relative to the critical case by use of the dimensionless quantity $\gamma > 0$:

$$
\frac{d}{S_{d-1}}\int\mathrm{d}\Omega_k\boldsymbol{u}_k\mathrm{i}\tilde{\Sigma}^K_{\alpha\beta}(k_{\mathrm{F}}\boldsymbol{u}_k) = \gamma\boldsymbol{\phi}_1\phi^2\tilde{F}^{\mathrm{corr}}_{\alpha\beta}(k_{\mathrm{F}})/\tau_{\mathrm{F}}\,. \tag{A.20}
$$

This justifies the RHS in the second line of the coupled PDEs in (37). $\gamma = 1$ corresponds to the critical case and we expect the near-critical regime to be described by $\gamma < 1$.

## B  Early-time exponential growth

In this Appendix, we solve the early-time regime of the coupled PDEs (40) in $d = 1$,

$$
\begin{cases}
\partial_\tau\phi + \partial_X\phi_1 = \phi(\phi^2-1)\,,\\
\partial_\tau\phi_1 + \partial_X\phi = \phi_1(\gamma\phi^2-1)\,,
\end{cases} \tag{B.1}
$$

with $0 < \gamma \leq 1$ and the initial conditions assumed to be $C^1$ at least,

$$
\begin{cases}
\phi(\tau=0,X) = 1-\delta\phi_0(X)\,,\text{ with }\delta\phi_0(|X|>R_0) = 0\,,\\
\phi_1(\tau=0,X) = 0\,.
\end{cases} \tag{B.2}
$$

Let us first introduce

$$
\begin{cases}
h := 1-\phi-\phi_1 = \delta\phi-\phi_1\,,\\
k := 1-\phi+\phi_1 = \delta\phi+\phi_1\,,
\end{cases} \tag{B.3}
$$

to obtain the linearized equations

$$\begin{cases} \partial_\tau h + \partial_X h = \dfrac{1+\gamma}{2}h + \dfrac{3-\gamma}{2}k\,, \\[2mm] \partial_\tau k - \partial_X k = \dfrac{1+\gamma}{2}k + \dfrac{3-\gamma}{2}h\,, \end{cases} \tag{B.4}$$

with initial condition $h(\tau=0,X)=k(\tau=0,X)=\delta\phi_0(X)$. Writing

$$\begin{cases} h(\tau,X)=e^{\frac{1+\gamma}{2}\tau}\Big[\delta\phi_0(X-\tau)+\dfrac{3-\gamma}{4}\int_{-\tau}^{\tau}ds\,\delta\phi_0(X+s)a(s,\tau)\Big], \\[2mm] k(\tau,X)=e^{\frac{1+\gamma}{2}\tau}\Big[\delta\phi_0(X+\tau)+\dfrac{3-\gamma}{4}\int_{-\tau}^{\tau}ds\,\delta\phi_0(X+s)b(s,\tau)\Big], \end{cases} \tag{B.5}$$

we obtain by direct substitution

$$\begin{cases} \partial_\tau h + \partial_X h - \dfrac{1+\gamma}{2}h = \dfrac{3-\gamma}{2}e^{\frac{1+\gamma}{2}\tau}\Big[a(\tau,\tau)\delta\phi_0(X+\tau)+\dfrac{1}{2}\int_{-\tau}^{\tau}ds\,\delta\phi_0(X+s)(\partial_\tau-\partial_s)a(s,\tau)\Big], \\[2mm] \partial_\tau k - \partial_X k - \dfrac{1+\gamma}{2}k = \dfrac{3-\gamma}{2}e^{\frac{1+\gamma}{2}\tau}\Big[b(-\tau,\tau)\delta\phi_0(X-\tau)+\dfrac{1}{2}\int_{-\tau}^{\tau}ds\,\delta\phi_0(X+s)(\partial_\tau+\partial_s)b(s,\tau)\Big], \end{cases} \tag{B.6}$$

so that the solution to Eq. (B.4) is obtained if

$$\begin{cases} a(\tau,\tau)=1\,, & (\partial_\tau-\partial_s)a(s,\tau)=\dfrac{3-\gamma}{2}b(s,\tau)\,, \\[2mm] b(-\tau,\tau)=1\,, & (\partial_\tau+\partial_s)b(s,\tau)=\dfrac{3-\gamma}{2}a(s,\tau)\,. \end{cases} \tag{B.7}$$

In turn, the solution to Eq. (B.7) is

$$\begin{cases} a(s,\tau)=I_0\Big(\dfrac{3-\gamma}{2}\sqrt{\tau^2-s^2}\Big)+I_1\Big(\dfrac{3-\gamma}{2}\sqrt{\tau^2-s^2}\Big)\sqrt{\dfrac{\tau-s}{\tau+s}}\,, \\[2mm] b(s,\tau)=I_0\Big(\dfrac{3-\gamma}{2}\sqrt{\tau^2-s^2}\Big)+I_1\Big(\dfrac{3-\gamma}{2}\sqrt{\tau^2-s^2}\Big)\sqrt{\dfrac{\tau+s}{\tau-s}}\,, \end{cases} \tag{B.8}$$

with $I_0$ and $I_1$ the modified Bessel functions of the first kind. Indeed, $a(\tau,\tau)=b(-\tau,\tau)=1$ since $I_0(0)=1$, and we check the differential equations in Eq. (B.7) hold. For the first one:

$$(\partial_\tau-\partial_s)a(s,\tau)=\dfrac{3-\gamma}{2}\left[\sqrt{\dfrac{\tau+s}{\tau-s}}I_0'\Big(\dfrac{3-\gamma}{2}\sqrt{\tau^2-s^2}\Big)+I_1'\Big(\dfrac{3-\gamma}{2}\sqrt{\tau^2-s^2}\Big)\right] \tag{B.9}$$

$$+\dfrac{I_1\Big(\frac{3-\gamma}{2}\sqrt{\tau^2-s^2}\Big)}{\sqrt{\tau^2-s^2}}$$

$$=\dfrac{3-\gamma}{2}b(s,\tau)\,, \tag{B.10}$$

where we used

$$I_0'(z)=I_1(z) \quad\text{and}\quad I_1'(z)=I_0(z)-\dfrac{1}{z}I_1(z)\,. \tag{B.11}$$

Similarly,

$$(\partial_\tau+\partial_s)b(s,\tau)=\dfrac{3-\gamma}{2}\left[\sqrt{\dfrac{\tau-s}{\tau+s}}I_0'\Big(\dfrac{3-\gamma}{2}\sqrt{\tau^2-s^2}\Big)+I_1'\Big(\dfrac{3-\gamma}{2}\sqrt{\tau^2-s^2}\Big)\right] \tag{B.12}$$

$$+\dfrac{I_1\Big(\frac{3-\gamma}{2}\sqrt{\tau^2-s^2}\Big)}{\sqrt{\tau^2-s^2}}$$

$$=\dfrac{3-\gamma}{2}a(s,\tau)\,. \tag{B.13}$$

Back in terms of $\delta\phi$, the exact solution of the linearized equations is

$$
\begin{aligned}
\delta\phi(\tau,X) &= \frac{h(\tau,X)+k(\tau,X)}{2} \\
&= e^{\frac{1+\gamma}{2}\tau}\Bigg[\frac{\delta\phi_0(X-\tau)+\delta\phi_0(X+\tau)}{2} \\
&\quad +\frac{3-\gamma}{4}\int_{-\tau}^{\tau}\mathrm{d}s\,\delta\phi_0(X+s)\bigg(I_0\Big(\frac{3-\gamma}{2}\sqrt{\tau^2-s^2}\Big)+I_1\Big(\frac{3-\gamma}{2}\sqrt{\tau^2-s^2}\Big)\frac{\tau}{\sqrt{\tau^2-s^2}}\bigg)\Bigg].
\end{aligned}
\tag{B.14}
$$

We now use that $\delta\phi_0$ is 0 except on a ball of radius $R_0$ around 0 (with $R_0$ of order 1, at most), and we call $\delta M_0 := \int \mathrm{d}X\,\delta\phi_0(X)$. The first terms in Eq. (B.14) vanish except around $X=\pm\tau$. The integrand is non-zero only around $s=-X$. Hence, for $\tau-|X|\gg 1$, we obtain

$$
\delta\phi(\tau,X)\simeq \delta M_0\frac{3-\gamma}{4}e^{\frac{1+\gamma}{2}\tau}\bigg(I_0\Big(\frac{3-\gamma}{2}\sqrt{\tau^2-X^2}\Big)+I_1\Big(\frac{3-\gamma}{2}\sqrt{\tau^2-X^2}\Big)\frac{\tau}{\sqrt{\tau^2-X^2}}\bigg).
\tag{B.15}
$$

Using

$$
I_0(z\gg 1)\simeq I_1(z\gg 1)\simeq \frac{e^z}{\sqrt{2\pi z}},
\tag{B.16}
$$

gives, for $\tau-|X|\gg 1$,

$$
\delta\phi(\tau,X)\simeq \delta M_0\frac{\sqrt{3-\gamma}}{2\sqrt{4\pi}(\tau^2-X^2)^{3/4}}e^{\frac{1+\gamma}{2}\tau+\frac{3-\gamma}{2}\sqrt{\tau^2-X^2}}\Big(\tau+\sqrt{\tau^2-X^2}\Big).
\tag{B.17}
$$

Recalling that $\sqrt{\tau^2-X^2}=\tau-X^2/(2\tau)+\mathcal{O}(X^4/\tau^3)$, we see in particular that for $\tau\gg 1$ and $|X|\ll\tau^{3/4}$, this yields the following form for the solution of the linear regime

$$
\delta\phi(\tau,X)\simeq \delta M_0\, e^{2\tau}\frac{e^{-(3-\gamma)\frac{X^2}{4\tau}}}{\sqrt{4\pi\tau/(3-\gamma)}},
\tag{B.18}
$$

which is valid as long as $\delta\phi(\tau,X)\ll 1$, *i.e.* for $\tau\ll\tau_* := -\frac{1}{2}\log\delta M_0$.

# C  Discontinuous front away from criticality ($\gamma\neq 1$)

In this Appendix, we compute the late-time solution of the following coupled PDEs

$$
\begin{cases}
\partial_\tau\phi+\partial_X\phi_1 = \phi(\phi^2-1), \\
\partial_\tau\phi_1+\partial_X\phi = \phi_1(\gamma\phi^2-1),
\end{cases}
\tag{C.1}
$$

for a generic value of the parameter $\gamma$. Let us assume that two symmetrical fronts, traveling at velocity $\pm 1$, develop at late times. We work in the reference frame of the right-moving front by using

$$
\begin{cases}
\phi_+(\tau,X) := \phi(\tau,X+\tau) \overset{\tau\to\infty}{\to} f_+(X), \\
\phi_{1+}(\tau,X) := \phi_1(\tau,X+\tau) \overset{\tau\to\infty}{\to} f_{1+}(X).
\end{cases}
\tag{C.2}
$$

They obey the equations

$$
\begin{cases}
\partial_\tau\phi_+ +\partial_X(\phi_{1+}-\phi_+) = \phi_+(\phi_+^2-1), \\
\partial_\tau\phi_{1+} -\partial_X(\phi_{1+}-\phi_+) = \phi_{1+}(\gamma\phi_+^2-1).
\end{cases}
\tag{C.3}
$$

As $\tau \to \infty$, this leads to

$$\partial_X(f_{1+} - f_+) = -f_+(1 - f_+^2) = f_{1+}\left(1 - \gamma f_+^2\right). \tag{C.4}$$

Hence

$$f_{1+} = -f_+ \frac{1 - f_+^2}{1 - \gamma f_+^2}, \tag{C.5}$$

and

$$\partial_X\left(f_+ \frac{1 - f_+^2}{1 - \gamma f_+^2} + f_+\right) = f_+\left(1 - f_+^2\right). \tag{C.6}$$

This is a first-order ODE that can be integrated by means of decomposition into simple fractions. Implicitly:

$$-\frac{2}{\gamma f_+^2 - 1} - \frac{(\gamma + 1)\log(1 - f_+^2)}{\gamma - 1} - \frac{(\gamma - 3)\log(1 - \gamma f_+^2)}{\gamma - 1} + 4\log f_+ = 2X + C. \tag{C.7}$$

The constant $C$ can be determined by solving $f_+(R_0^-)$ (*i.e.* the discontinuity). We may extract the missing information from "the other side", *i.e.* on the left-moving front. Introduce

$$\phi_-(\tau, X) = \phi(\tau, X - \tau), \qquad \phi_{1-}(\tau, X) = \phi_1(\tau, X - \tau). \tag{C.8}$$

Using $\partial_\tau \phi_- = \partial_\tau \phi - \partial_X \phi$, we have

$$\begin{cases} \partial_\tau \phi_- + \partial_X(\phi_{1-} + \phi_-) = -\phi_-(1 - \phi_-^2), \\ \partial_\tau \phi_{1-} + \partial_X(\phi_{1-} + \phi_-) = -\phi_{1-}(1 - \gamma \phi_-^2). \end{cases} \tag{C.9}$$

Then, integrating the first equation:

$$\phi_-(\tau, X) = e^{-\int_0^\tau ds\,(1 - \phi_-(s,X)^2)}\left[\phi_0(X) - \int_0^\tau du\, e^{\int_0^u ds\,(1 - \phi_-(s,X)^2)}(\partial_X \phi_{1-} + \partial_X \phi_-)(u, X)\right] \tag{C.10}$$

$$= e^{-\int_0^\tau ds\,(1 - \phi_-(s,X)^2)}\phi_0(X) - \int_0^\tau du\, e^{-\int_u^\tau ds\,(1 - \phi_-(s,X)^2)}(\partial_X \phi_{1-} + \partial_X \phi_-)(u, X) \tag{C.11}$$

$$= e^{-\int_0^\tau ds\,(1 - \phi_-(\tau - s,X)^2)}\phi_0(X)$$
$$- \int_0^\tau du\, e^{-\int_0^u ds\,(1 - \phi_-(\tau - s,X)^2)}(\partial_X \phi_{1-} + \partial_X \phi_-)(\tau - u, X). \tag{C.12}$$

Add $2\tau$ to $X$, and use $\phi_-(\tau, X + 2\tau) = \phi_+(\tau, X)$:

$$\phi_+(\tau, X) = e^{-\int_0^\tau ds\,(1 - \phi_+(\tau - s,X + 2s)^2)}\phi_0(X + 2\tau)$$
$$- \int_0^\tau du\, e^{-\int_0^u ds\,(1 - \phi_+(\tau - s,X + 2s)^2)}(\partial_X \phi_{1+} + \partial_X \phi_+)(\tau - u, X + 2u). \tag{C.13}$$

When $X > R_0$, recalling that we postulate a solution such that $\phi_+(\tau, X > R_0) = 1$, the above equation trivially simplifies to $1 = 1 - 0$. When $X < R_0$, we take $\tau > \frac{R_0 - X}{2}$. Then, since $\phi_+(\tau, X > R_0) = 1$, we have:

$$\phi_+(\tau, X) = e^{-\int_0^{\frac{R_0 - X}{2}} ds\,(1 - \phi_+(\tau - s,X + 2s)^2)}$$
$$- \int_0^{\frac{R_0 - X}{2}} du\, e^{-\int_0^u ds\,(1 - \phi_+(\tau - s,X + 2s)^2)}(\partial_X \phi_{+1} + \partial_X \phi_+)(\tau - u, X + 2u). \tag{C.14}$$

Before taking the limit $\tau \to \infty$, one is to be careful because the spatial derivatives in the integrand above are unbounded as we expect the discontinuity to develop: one cannot blindly replace $\phi_+$ by $f_+$. Instead, write:

$$
\begin{aligned}
(\partial_X \phi_{1+} + \partial_X \phi_+)(\tau - u, X + 2u) &= \frac{1}{2}\frac{\mathrm{d}}{\mathrm{d}u}\Big[(\phi_{1+} + \phi_+)(\tau - u, X + 2u)\Big] \\
&\quad + \frac{1}{2}(\partial_\tau \phi_{1+} + \partial_\tau \phi_+)(\tau - u, X + 2u) \\
&= \frac{1}{2}\frac{\mathrm{d}}{\mathrm{d}u}\Big[(\phi_{1+} + \phi_+)(\tau - u, X + 2u)\Big] \\
&\quad - \frac{1}{2}\Big[\phi_+(1 - \phi_+^2) + \phi_{1+}(1 - \gamma\phi_+^2)\Big](\tau - u, X + 2u),
\end{aligned}
$$
(C.15)

using Eq. (C.3). Inserting Eq. (C.15) into Eq. (C.14) and integrate by parts:

$$
\begin{aligned}
\phi_+(\tau, X) &= \mathrm{e}^{-\int_0^{\frac{R_0 - X}{2}} \mathrm{d}s\,(\cdots)} - \frac{1}{2}(\phi_{1+} + \phi_+)\left(\tau - \frac{R_0 - X}{2}, R_0\right)\mathrm{e}^{-\int_0^{\frac{R_0 - X}{2}} \mathrm{d}s\,(\cdots)} \\
&\quad + \frac{1}{2}(\phi_{1+} + \phi_+)(\tau, X) + \int_0^{\frac{R_0 - X}{2}} \mathrm{d}u\,(\cdots).
\end{aligned}
$$
(C.16)

Crucially, the terms collected in $(\cdots)$ are bounded. Moreover, $\phi_{1+}$ and $\phi_+$ are continuous and $(\phi_{1+} + \phi_+)(\tau, R_0) = 1$ for any finite time $\tau$. Therefore, we can replace $(\phi_{1+} + \phi_+)(\tau - \frac{R_0 - X}{2}, R_0)$ by 1 all times. Later sending $\tau \to \infty$, we obtain

$$
f_+(X) = \mathrm{e}^{-\int_0^{\frac{R_0 - X}{2}} \mathrm{d}s\,(\cdots)} - \frac{1}{2}\mathrm{e}^{-\int_0^{\frac{R_0 - X}{2}} \mathrm{d}s\,(\cdots)} + \frac{1}{2}(f_{1+} + f_+)(X) + \int_0^{\frac{R_0 - X}{2}} \mathrm{d}u\,(\cdots),
$$
(C.17)

and send $X \to R_0^-$

$$
f_+(R_0^-) = \frac{1}{2} + \frac{1}{2}(f_{1+} + f_+)(R_0^-) \qquad i.e. \qquad f_{1+}(R_0^-) = f_+(R_0^-) - 1.
$$
(C.18)

Let us call $L := f_+(R_0^-)$. Using Eq. (C.5), we have

$$
-L\frac{1 - L^2}{1 - \gamma L^2} = L - 1.
$$
(C.19)

Simplifying by $L - 1$

$$
(1 + \gamma)L^2 + L - 1 = 0.
$$
(C.20)

The positive solution is

$$
L(\gamma) = \frac{\sqrt{5 + 4\gamma} - 1}{2(1 + \gamma)}.
$$
(C.21)

This quantity monotonically interpolates from $L = 1/2$ for $\gamma = 1$ to $L = 1/(\text{golden ratio})$ for $\gamma = 0$.

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
