# Peer review of "Traveling discontinuity at the quantum butterfly front"

_SciPost Physics, doi:SciPost Phys. 15, 042 (2023)_

## Round 1 · Referee Report · Anonymous (Referee 1) · 2023-2-15

Report

The authors study a standard probe of scrambling, the so-called out-of-time-ordered correlators (OTOCs) in a model of interacting electrons in the vicinity of a superconducting phase transitions. Within a "many-world", or augmented, Keldysh formalism, they derive a kinetic equation for the so-called inter-world distribution function. The authors then proceed to simplify the latter, obtaining an effective description of the scrambling dynamics in terms of a set of coupled PDEs, which allow for an analytic solution.

From the physical point of view, the authors describe in detail the geometric structure of the OTOC wavefront. Most prominently, they find that the propagating wavefront develops a sharp discontinuity at late time, without any diffusive broadening of the light-cone and with no exponentially decaying tails outside of it. Although this is different from what happens in other models, it is not possible to rule out that this is an effect of the performed approximations, as the authors also discuss. In any case, I find the description of the OTOC wavefront very interesting.

I believe the paper is extremely well written. The motivations and framework are very clear. In addition, while the computations are very technical (and I am not an expert in the specific techniques used), I believe the main logic of the derivations is always easy to grasp. In addition, it is always stated very clearly what the approximations are, and how they could affect the final result (including, for instance, the sharp discontinuity of the OTOC wavefront).

Finally, the topic is certainly timely and of broad interest. For these reasons, I believe this is an excellent submission and I recommend publication of the paper, essentially as is.
  • validity: -
  • significance: -
  • originality: -
  • clarity: -
  • formatting: -
  • grammar: -

Author:  Camille Aron  on 2023-05-08  [id 3651]

(in reply to Report 1 on 2023-02-15)

We would like to express our gratitude to the Referee for his/her time and effort in reviewing our submission and for his/her appreciation of our work. We have revised our manuscript following the suggestions of Referee 2. The main modifications are highlighted in red for ease of reference. We also added a novel Appendix B which provides a rigorous computation of the scrambling dynamics in the early-time regime.

---

## Round 1 · Referee Report · Anonymous (Referee 2) · 2023-2-18

Report

The authors studied the out-of-time-ordered correlation function of interacting electrons in the vicinity of the superconductivity phase transition on the disordered side. They used the inter-world kinetic theory on the Keldysh contour to derive a set of coupled differential equations describing information scrambling in the system. This set of coupled differential equations differs from the FKPP equation derived before. The authors found that the equations display a traveling wave solution, and the asymptotic wavefront develops a discontinuity at the wavefront, which is absent in the conventional FKPP equation. The authors also remarked on the possible reasons for the discontinuity, which they left for future study.

The result of the work is interesting because the derived coupled differential equation from the partial wave expansion displays an elegant form and results in interesting wavefront dynamics that have not been studied before. Therefore, the result is worth publishing.

Here are some suggestions for the authors to consider:

  1. The authors should explain the role of superconductivity fluctuations and how the vicinity of the phase transition affects the calculation in the introduction. It would also be helpful to explain how the equation would change if the system were far from the phase transition and whether any phase transition fluctuations would lead to the same equation.

  2. The authors should relate the inter-world distribution function to OTOC more closely in the summary and main results section. They should also explain how to calculate OTOC from the inter-world distribution function.

  3. The calculation was performed at a finite temperature, so the authors should explain how temperature enters the equation.

  4. The authors should comment on possible numerical simulations needed to verify the discontinuity in the wavefront.

  • validity: high
  • significance: high
  • originality: top
  • clarity: good
  • formatting: excellent
  • grammar: perfect

Author:  Camille Aron  on 2023-05-08  [id 3650]

(in reply to Report 2 on 2023-02-18)

We would like to express our gratitude to the Referee for his/her time, appreciation of our work, and for providing valuable feedback and insightful suggestions for improvement.

In response to his/her suggestions, we have carefully revised our manuscript, highlighting the main modifications in red for ease of reference. We also added a novel Appendix B which provides a rigorous computation of the scrambling dynamics in the early-time regime. Please find below our answers to each point raised by the Referee.

  1. We added explanations to our introduction: "The long wavelength fluctuations close to criticality produce a separation of scales which we leverage to derive analytic results. How the results get modified on moving away from criticality is transparent in our derivation and our approach can be systematized to address other near-critical quantum many-body systems."

  2. Following the Referee's advice, we have now included the explicit relation between OTOC and the interworld distribution function around Eq. (16). For a more detailed construction of the formalism, the reader is also referred to Ref. [15] (I. L. Aleiner, L. Faoro and L. B. Ioffe, Microscopic model of quantum butterfly effect: Out-of-time-order correlators and traveling combustion waves, Annals of Physics 375, 378 (2016) for a more detailed description of the formalism.)

  3. As usual in condensed matter systems at or below room temperature, the temperature is much smaller than the Fermi energy at the eV scale. Temperature enters in the parameters of our resulting model given by the equations (36): in the inelastic scattering rate 1/\tau_F, as well as in the `distance' to criticality \gamma. We have now stated it explicitly below Eq. (37).

  4. We added a comment on the possible numerical checks of the discontinuous front in Sect. 6 "Discussion and conclusion". Verifying the shock-wave dynamics of scrambling by means of direct numerical simulations of a microscopic Hamiltonian system is expected to be a difficult task. Indeed, the scrambling front propagates fast and the discontinuity only develops at late times. This means that one should simulate rather large systems for a long time to ensure that the light ray from the initial perturbation at x=0 has not reached the boundary of the system before the onset of the discontinuous front is clearly visible. Tensor network approaches in 1d may be suited to meet the challenge.

---

## Editorial Decision

published